# From rapid visual survey to multi-hazard risk prioritisation and numerical fragility of school buildings

Roberto Gentile[1], Carmine Galasso[2], Yunita Idris[3], Ibnu Rusydy[3], Ella Meilianda[3]

[1]Institute for Risk and Disaster Reduction, University College London, London, United Kingdom
[2]Department of Civil, Environmental and Geomatic Engineering, University College London, London, United Kingdom
[3]Tsunami and Disaster Mitigation Research Center (TDMRC), Universitas Syiah Kuala, Banda Aceh, Indonesia

*Correspondence to*: Roberto Gentile (r.gentile@ucl.ac.uk)

**Abstract.** Regional seismic risk assessment is paramount in earthquake-prone areas, for instance, to define and implement prioritisation schemes for earthquake risk reduction. As part of the *INdonesia School Programme to Increase REsilience*
(INSPIRE), this paper proposes an ad-hoc rapid visual survey form, allowing to 1) calculate the newly-proposed INSPIRE seismic risk prioritisation index, which is an empirical proxy for the relative seismic risk of reinforced concrete (RC) buildings within a given building portfolio; 2) calculate the Papathoma Tsunami Vulnerability Assessment (PTVA) index, in any of its variations; 3) define one or more archetype buildings representative of the analysed portfolio; 4) derive detailed numerical models of the archetype buildings, provided that simulated design is used to cross-check the model assumptions. The proposed
INSPIRE index combines a baseline score, calibrated based on fragility curves, and a performance modifier, calibrated through the Analytic Hierarchy Process (AHP) to minimise subjectivity. An attempt to define a multi-hazard prioritisation scheme is proposed, combining the INSPIRE and PTVA indices. Such a multi-level framework is implemented for 85 RC school buildings in Banda Aceh, Indonesia, the mostly affected city by the 2004 Indian Ocean earthquake-tsunami sequence. As part of the proposed framework, two archetype buildings representative of the entire portfolio are defined based on the collected
data. Their seismic performance is analysed by means of non-linear static analyses, using both the analytical Simple Lateral Mechanism Analysis (SLaMA) method and numerical finite element pushover analyses to investigate the expected plastic mechanisms and derive displacement/drift thresholds to define appropriate damage states. Finally, non-linear dynamic analyses are performed to derive fragility curves for the archetype buildings. This paper demonstrates the effectiveness of the INSPIRE data collection form and proposed index in providing a rational method to derive seismic risk prioritisation schemes and in
allowing the definition of archetype buildings for more detailed evaluations/analyses.

## 1. Introduction

Regional seismic risk assessment is paramount in highly earthquake-prone areas. In fact, in several countries around the world, a large portion of the building stock has been designed/constructed according to obsolete structural codes, which include little-to-no provisions for earthquake resistance and detailing. Several Reinforced Concrete (RC) buildings fall in this category, and
they often represent the highest share for both residential and commercial occupancy in many countries (e.g., in Italy 48% of

the buildings constructed after 1971 is made of RC; ISTAT 2011). RC structural systems are also widely used in the design of critical infrastructure, such as hospitals and school facilities. Those are the focus of this paper. Clearly, it is desirable that any risk-mitigation strategy designed by governmental agencies should be based on a rational understanding of the risk of large building groups – or portfolios – at a country level (or in a smaller region). However, it is cost-ineffective to perform detailed

structural simulations for a large amount of structures, given the shortage of both financial and technical/computational resources. Therefore, a multi-level approach is usually preferred, starting with a screening based on simplified and rapid methods and performing more detailed structural analyses only for selected groups of structures at higher relative risk and for which an archetype (or index) building can be identified (e.g., FEMA P-154, 2015, Benedetti and Petrini, 1984, Grant et al., 2007).

Common approaches for regional seismic risk assessment of RC buildings (see *Section 2* for more details) refer to typological approaches based on pre-determined building categories (e.g., Giovinazzi and Lagomarsino, 2004), or the use of Rapid Visual Survey (RVS) forms and calibrated empirical seismic vulnerability/risk indices (e.g., Uva et al., 2013). Although these approaches rely on various assumptions and usually involve some degree of subjectivity, such simplified methods provide valuable proxies to develop prioritisation schemes (i.e., performing a ranking of the buildings in a given portfolio based on

their relative vulnerability or risk-related metrics). As discussed, such simplified methods include some degree of subjectivity by the analyst, mainly reflected in the choice and the assigned relative importance of the parameters involved in the analysis. Moreover, given the low amount of information required, such methods do not allow to further refine the analysis, providing a more detailed, second-level seismic risk assessment. Finally, those methods mostly refer to seismic hazard, which in some countries might not be enough for the development of a rational multi-hazard prioritisation scheme.

Numerous evidences of previous natural hazard events have highlighted the vulnerability of school infrastructure to natural hazards and particularly to earthquake-induced ground shaking. From the structural and architectural points of view, school buildings are especially vulnerable given structural characteristics that typically include large rooms, large windows (particularly in tropical climates), and corridors, all of which may represent seismic vulnerability factors. At the same time, schools play a critical role in the education of a community's next generation, with school children being one of the most

vulnerable components of the society due to their age and their developmental stage. A safer and resilient school can save valuable lives of children and help to bring normality back to society in times of disaster. These considerations set school buildings apart from their peers in terms of priority for assessment and resource allocation for structural retrofitting. In fact, some of the world disaster reduction campaigns by the United Nations International Strategy for Disaster Reduction (UNISDR) were carried out together with various partner organizations under the theme of "Disaster Risk Reduction Begins at School"

(UN, 2009). Recently, the Comprehensive School Safety Framework (CSSF; GADRRRES, 2017) has proposed an integrated approach to reduce disaster risk and promote resilience in the education sector. The CSSF is funded on three pillars: "*Safe Learning Facilities*" (including "*implementing assessment and prioritisation plans for retrofitting or replacing unsafe schools, including relocation*"), "*School Disaster Management*", and "*Risk Reduction and Resilience Education*".

Based on the above discussion, a new RVS form and a seismic risk prioritisation index for RC buildings are proposed in this study to address the above-mentioned gaps. Both the RVS form and the seismic index are the first outcomes of the INSPIRE project (*INdonesia School Programme to Increase REsilience*). INSPIRE looks to develop an advanced, harmonised and science-based risk assessment framework for school infrastructure in Indonesia subjected to cascading earthquake-tsunami hazards. It also assesses the effectiveness of different soft (e.g., risk reduction education) and hard (e.g., retrofitting) mitigation measures in reducing casualties, economic loss and disruption to school infrastructure, thus increasing community resilience. The INSPIRE RVS form (Figure 1) is designed to be completed by trained engineers in approximately 20-30 minutes - depending on the size of the building - by means of a sidewalk survey. This is a one-page form including various sections related to the general identification and geolocation of the building, its geometric properties (including space for sketching the building's shape and footprint), and its structural characteristics and deficiencies, including the structural typology and the dimensions/details of the main structural members. It is also possible to assign a "confidence level" for each parameter, allowing for a better classification and weighting of the data after a campaign of RVSs. The back of the form is used to provide definition of both the parameters and the confidence levels and provides blank space that be used to register extra information. The collected data is fully compatible with both the Global Earthquake Model (GEM) Building Taxonomy (Brzev et al., 2013) and the Hazard United States (HAZUS) model (Kircher et al., 2006). The collected information allows to 1) calculate the proposed INSPIRE seismic risk prioritisation index, introduced in *Section 3*; 2) calculate the Papathoma Tsunami Vulnerability Assessment index (PTVA4, Dall'Osso et al., 2016, in any of its variations, described in *Section 4.2*); 3) define one or more representative archetype buildings consistent with the local building codes and practice; 4) derive detailed numerical models of the archetype buildings, provided that simulated design is used to cross-check the model assumptions. The unique features of the INSPIRE form include: 1) the possibility of considering both seismic and tsunami hazards while requiring a reasonable amount of time to complete the survey; 2) the consideration of "confidence levels" for the parameters, which is particularly useful in deriving statistics; 3) the possibility to be expanded to include other hazards with simple modifications/customisations.

In particular, the INSPIRE seismic risk prioritisation index (*Section 3*) aims at providing a simple method to derive a prioritisation scheme, minimising the subjectivity involved in the calculation. In fact, mechanics-based fragility functions are used to define a baseline score. A performance modifier is defined based on parameters that can jeopardise the seismic performance of a building (e.g., presence of short columns, pounding potential). The weight assigned to each parameter is defined through the Analytical Hierarchy Process (AHP; Saaty, 1980), providing a mathematically-consistent and rational solution to the weighting process.

In this study, the INSPIRE RVS form and proposed multi-hazard risk prioritisation index are applied to a portfolio of 85 RC school buildings in Banda Aceh, Indonesia, highlighting the simplicity and rapidity of the whole process. Moreover, consistently with the proposed multi-level framework, a detailed analytical and numerical seismic fragility assessment is provided for the identified archetype school building, demonstrating the effectiveness of the INSPIRE RVS form in providing inputs and allowing more detailed analyses.

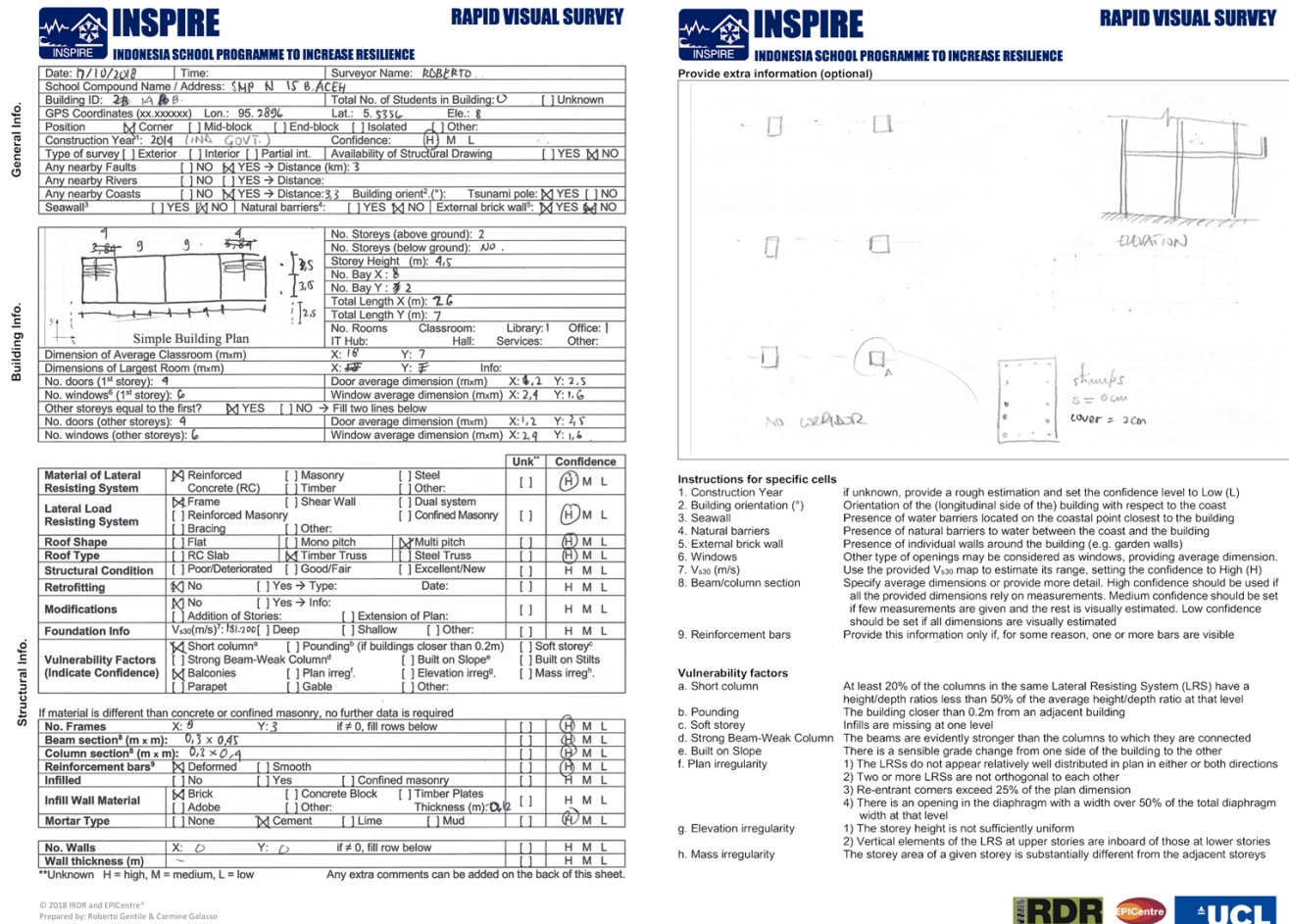

**Figure 1 INSPIRE Rapid Visual Survey form completed for building with ID 14A (*Section* 4.1).**

## 2. Seismic risk prioritisation schemes: a review

Various methodologies to derive prioritisation schemes for buildings based on their relative seismic vulnerability/risk are available in the scientific literature and or international standards/guidelines. Each of these is characterised by a different underlying approach, basic assumptions and/or applicability conditions. Most of these are based on the calculation of a seismic vulnerability/risk index for a portfolio of buildings. The results can be used to rank those building, defining a priority list of structures requiring further investigation. A comprehensive overview of this previous research is not within the scope of this paper; however, a briefing of some relevant past initiatives is presented here.

The procedure proposed in the guidelines by the Applied Technology Council (ATC 3-06, 1978) uses a strength-based approach to define an earthquake capacity ratio, comparing the 'actual' strength of the building to the code requirement for new ones. Adjustments are also adopted to consider in-situ material properties and insufficient detailing (compared to modern

design). Such capacity-to-demand ratio is defined as the earthquake capacity ratio, and it is calculated as the minimum of the component-by-component strength ratios.

The Alaska's Department of Education (1997), among others, has produced surveying forms to assess the structural conditions of buildings and the associated seismic vulnerability, with focus on school buildings. Such forms mainly consist of checklists investigating areas of potential concerns for seismic vulnerability. On the other hand, the procedure introduced by the Federal Emergency Management Agency (FEMA P-154, 2015) is based on a Rapid Visual Screening of buildings and a two-level approach for a fast assignment of a seismic vulnerability index (which requires no mechanical calculation from the user). The companion FEMA P-155, 2015 describes the rationale behind the scoring system, which is directly connected to the probability of collapse of archetype building categories. Such method is based on the HAZUS framework (and typological force-displacement curves) to define the building categories and to derive a seismic-only assessment.

The New Zealand Society for Earthquake Engineering (NZSEE) defines an Initial Evaluation Procedure (IEP) in the NZSEE guidelines of 2017, providing a broad indication of the "seismic rating" of a building based on a sidewalk survey. The score is expressed in terms of the percentage of New Building Standard (%NBS), which is the ratio of the displacement capacity of the building for the Life Safety limit state, over the minimum capacity required for a new building for the same limit state. Firstly, a baseline %NBS is calculated using specifically-tabulated coefficients relating to year of design, strengthening interventions, importance of the structure, assumed ductility capacity, site hazard, presence of near-fault effects, soil type, etc. It is assumed that the capacity of the building cannot be lower than the minimum specified by the code valid for the year of design. This is only true if building codes are legally enforced. If this is not the case, the seismic vulnerability assessment may be characterised by a higher level of epistemic uncertainty. The baseline value is reduced if structural weaknesses are present (e.g., pounding potential, soil characteristics, presence of vertical and plan irregularity, presence of short columns).

The Italian National Group for Earthquake Defence (*Gruppo Nazionale per la Difesa dai Terremoti*, GNDT) has also provided a seismic vulnerability index (Benedetti and Petrini, 1984, Angeletti et al., 1988) based on simple assessment forms including, among other parameters, the structural material, the typology of the Lateral-Load Resisting System (LLRS), the quality of the building materials and the overall construction, and the existing damage level (if any). The vulnerability index is based on a weighted sum of such parameters and is defined in the range [0, 100] for masonry and [−25, 100] for reinforced concrete, mostly based on expert opinion. A higher value of the index indicates a higher seismic vulnerability. Interestingly, the vulnerability index by GNDT has been correlated with structural damage in past earthquakes (Grimaz et al., 1996, Zonno et al., 1999), allowing to indirectly calculate the PGA which is likely to induce collapse.

Other rapid surveying forms and rapid procedures have been proposed by different authorities and organisations, such as the World Health Organisation (WHO) and the United Nations (UN), with special focus on developing countries. For instance, Dhungel et al., 2012 collected and assessed the physical condition of 1,381 school building units in Nepal. The data was collected by mobilising the school teachers; school vulnerability, calculated on the basis on the empirical weightage on different factors (e.g. structural material, number of storeys, shape of the roof), was used to estimate the possible damage/casualties/injuries for earthquakes of different seismic intensities.

Finally, a broader perspective is provided in the work by Grant et al., 2007 which proposes a four-level prioritisation framework focused on school buildings, filtering the buildings with increased level of detail, allowing the policy-makers to choose the filtering thresholds based on the available resources. The idea is to firstly check the code-based demand deficit of the buildings, comparing the new structural code demand to the code appropriate for the year of construction, in terms of PGA. For the buildings with a deficit above a given threshold, the GNDT index is calculated. The buildings with the highest rating are assessed with a simplified mechanics-based procedure, and finally the most critical ones are assessed with structural models providing full details.

## 3. Definition of the INSPIRE seismic risk index

The INSPIRE index ($I_V$) is an empirical proxy for the relative seismic risk of various buildings within a given building portfolio. The index is specifically defined for RC buildings, although its definition can be extended to other building types, and it consists of two parts: the baseline score ($I_{BL}$) and a performance modifier ($\Delta I_{PM}$), which are summed up to obtain the final index (Eq. 1). In its current version, the baseline score is based on the fragility curves available in the HAZUS MH4 framework (Kircher et al., 2006), which allows to have a transparent and consistent fragility estimation for a wide range of structural typologies. HAZUS fragilities are defined by three "primary" parameters: RC Basic Structural System (BSS: Frame, Infilled frame or Wall), building height (Low-rise, Mid-rise or High-rise) and seismic design criteria (Pre-, Low-, Moderate- or High-code). On the other hand, the performance modifier is based on the score of the building regarding eight "secondary" parameters (preservation condition, plan shape, storey height uniformity, added storeys, infills at ground storey, short columns, pounding, unfavourable soil), which, if present, are deemed to jeopardise the performance of the building. It is worth mentioning that the selection of the appropriate fragility curves (HAZUS category) for each building in the portfolio is an expert decision provided by the analyst. However, as further discussed below, any other type of fragility curves, if deemed appropriate, can be used to re-define the INSPIRE index according to the proposed methodology.

$$I_V = I_{BL} + \Delta I_{PM}$$

### 3.1. Baseline score

The HAZUS MH4 framework provide, among other information, an extensive set of fragility curves representing the seismic performance of archetype buildings which are classified based on four parameters: material (*Mat*), BSS, building *Height* and seismic *Code* level (defined according to the Uniform Building Code 1994, UBC 1994). Adopting the HAZUS fragility database as a reference is further motivated by the fact that several countries around the world have adopted seismic provisions which are, to different extents, consistent with the recommendations of the UBC 1994. For each archetype building category, four fragility functions are provided in HAZUS, respectively for the Slight, Moderate, Extensive and Complete Damage States (DS), or limit states. The fragility functions are defined as lognormal Cumulative Distribution Functions, or CDF (Eq. 2), in

terms of different Intensity Measures, among which the Peak Ground Acceleration (PGA). The curves are defined by a median PGA ($\mu$) and a dispersion term ($\beta$).

$$P(DS \geq DS_i | Mat, BSS, Code, Height, PGA) = \Phi\left(\frac{\ln(PGA/\mu)}{\beta}\right), i = 1:4 \qquad\qquad 2$$

For the scope of this paper, the HAZUS fragility database has been filtered to consider only the curves related to RC buildings (namely, categories C1, C2, C3). Moreover, only the fragility curves corresponding to the Extensive Damage limit state (DS3) have been considered, which are mainly related to the Life Safety performance objective according to modern seismic codes (e.g., ASCE 7-10, Eurocode 8, NZSEE 2017). The selected fragility curves are reported in Table 1. Details of the involved parameters is provided in Table 2.

**Table 1 Selected fragility curves from HAZUS MH4 framework (Kircher et al., 2006).**

| HAZUS Basic Structural System | Code Level | Height | $\mu$: Median PGA [g] | $\beta$: Dispersion | Inter-storey drift limit for DS3 [Rad] |
|---|---|---|---|---|---|
| C1 Concrete Moment Frame | Pre Code | Low Rise | 0.21 | 0.64 | 0.016 |
| | | Mid Rise | 0.26 | 0.64 | 0.011 |
| | | High Rise | 0.21 | 0.64 | 0.008 |
| | Low Code | Low Rise | 0.27 | 0.64 | 0.020 |
| | | Mid Rise | 0.32 | 0.64 | 0.013 |
| | | High Rise | 0.27 | 0.64 | 0.010 |
| | Mod Code | Low Rise | 0.41 | 0.64 | 0.023 |
| | | Mid Rise | 0.49 | 0.64 | 0.015 |
| | | High Rise | 0.41 | 0.64 | 0.011 |
| | High Code | Low Rise | 0.70 | 0.64 | 0.030 |
| | | Mid Rise | 0.73 | 0.64 | 0.020 |
| | | High Rise | 0.62 | 0.64 | 0.015 |
| C2 Concrete Shear Wall | Pre Code | Low Rise | 0.24 | 0.64 | 0.016 |
| | | Mid Rise | 0.30 | 0.64 | 0.011 |
| | | High Rise | 0.31 | 0.64 | 0.008 |
| | Low Code | Low Rise | 0.30 | 0.64 | 0.020 |
| | | Mid Rise | 0.38 | 0.64 | 0.013 |
| | | High Rise | 0.38 | 0.64 | 0.010 |
| | Mod Code | Low Rise | 0.49 | 0.64 | 0.023 |
| | | Mid Rise | 0.55 | 0.64 | 0.015 |
| | | High Rise | 0.57 | 0.64 | 0.011 |
| | High Code | Low Rise | 0.90 | 0.64 | 0.030 |
| | | Mid Rise | 0.87 | 0.64 | 0.020 |
| | | High Rise | 0.82 | 0.64 | 0.015 |
| C3 Concrete Infilled Frame | Pre Code | Low Rise | 0.21 | 0.64 | 0.012 |
| | | Mid Rise | 0.25 | 0.64 | 0.008 |
| | | High Rise | 0.27 | 0.64 | 0.006 |
| | Low Code | Low Rise | 0.26 | 0.64 | 0.015 |
| | | Mid Rise | 0.32 | 0.64 | 0.010 |
| | | High Rise | 0.33 | 0.64 | 0.007 |
| | Mod Code | Low Rise | n.a. | n.a. | n.a. |
| | | Mid Rise | n.a. | n.a. | n.a. |
| | | High Rise | n.a. | n.a. | n.a. |
| | High Code | Low Rise | n.a. | n.a. | n.a. |
| | | Mid Rise | n.a. | n.a. | n.a. |
| | | High Rise | n.a. | n.a. | n.a. |

**Table 2 Description of the HAZUS MH4 categories involved in the INSPIRE index (modified after Kircher et al., 2006).**

| | Description |
|---|---|
| C1 Concrete Moment Frame | These buildings have a frame of reinforced concrete columns and beams. Some older concrete frames may be proportioned and detailed such that brittle failure of the frame members can occur in earthquakes leading to partial or full collapse of the buildings. Modern frames in zones of high seismicity are proportioned and detailed for ductile behaviour and are likely to undergo large deformations during an earthquake without brittle failure of frame members and collapse. |
| C2 Concrete Shear Wall | The vertical components of the lateral-force-resisting system in these buildings are concrete shear walls that are usually bearing walls. In older buildings, the walls often are quite extensive and the wall stresses are low but reinforcing is light. In newer buildings, the shear walls often are limited in extent, generating concerns about boundary members and overturning forces. |
| C3 Concrete Infilled Frame | These buildings are made of reinforced concrete columns and beams and unreinforced masonry infill walls. In these buildings, the shear strength of the columns, after cracking of the infill, may limit the semi-ductile behaviour of the system. |
| Building height for C1, C2, C3 | Low Rise        1:3 storeys <br> Mid Rise        4:7 storeys <br> High Rise       8+ storeys |
| Damage State 3 (DS3): Extensive Damage [C1] | Some of the frame elements have reached their ultimate capacity indicated in ductile frames by large flexural cracks, spalled concrete and buckled main reinforcement; nonductile frame elements may have suffered shear failures or bond failures at reinforcement splices, or broken ties or buckled main reinforcement in columns which may result in partial collapse. |
| Damage State 3 (DS3): Extensive Damage [C2] | Most concrete shear walls have exceeded their yield capacities; some walls have exceeded their ultimate capacities indicated by large, through-the-wall diagonal cracks, extensive spalling around the cracks and visibly buckled wall reinforcement or rotation of narrow walls with inadequate foundations. Partial collapse may occur due to failure of nonductile columns not designed to resist lateral loads. |
| Damage State 3 (DS3): Extensive Damage [C3] | Most infill walls exhibit large cracks; some bricks may dislodge and fall; some infill walls may bulge out-of-plane; few walls may fall partially or fully; few concrete columns or beams may fail in shear resulting in partial collapse. Structure may exhibit permanent lateral deformation. |
| Pre Code | Gravity-dominated structures. No seismic design/detailing is available. Such structures are likely built prior to the introduction of seismic codes. |
| Low Code | According to the provisions in UBC1994 (seismic zone 1), such buildings can sustain a base shear at most equal to 7.5% of the total weight (assuming an elastic design, no importance factor and stiff soil). The real lateral capacity is likely to be lower than this maximum value. |
| Moderate Code | According to the provisions in UBC1994 (seismic zone 2b), such buildings can sustain a base shear at most equal to 20% of the total weight (assuming an elastic design, no importance factor and stiff soil). The real lateral capacity is likely to be lower than this maximum value. |
| High Code | According to the provisions in UBC1994 (seismic zone 4), such buildings can sustain a base shear at most equal to 40% of the total weight (assuming an elastic design, no importance factor and stiff soil). The real lateral capacity is likely to be lower than this maximum value. |

To define the baseline score of the INSPIRE index, for each considered archetype building category, and its corresponding fragility curve, the probability to exceed DS3 (Extensive Damage) is calculated (Figure 2a) for three levels of PGA, 0.1g, 0.25g, 0.4g, respectively corresponding to low, moderate and high seismicity levels. The analyst will select the seismicity level appropriate for the considered building portfolio/geographic area. It is worth noting that, in modern seismic codes, DS3 is related to the Life Safety performance objective. The process above is repeated for each archetype building category in the

HAZUS database, such that it is possible to map the building basic parameters to the exceeding probability of the DS, conditional to the considered PGA value ($P_{HAZUS} = P(DS \geq DS_3 | Mat, BSS, Code, Height, PGA)$. The baseline score of the risk index is set to be proportional to such exceeding probability, after a re-scaling in a range [1%, 50%] based on the minimum/maximum DS3 exceeding probability in the complete (non-filtered) HAZUS database and calculated according to Eq. 3. In such equation, $P_{HAZUS,max}$ and $P_{HAZUS,min}$ are the maximum and minimum DS3 exceeding probability in the HAZUS database for the chosen levels of PGA (indicated with dots in Figure 2b), while $P_{HAZUS}$ is the DS3 exceeding probability of the considered building, for the chosen level of PGA.

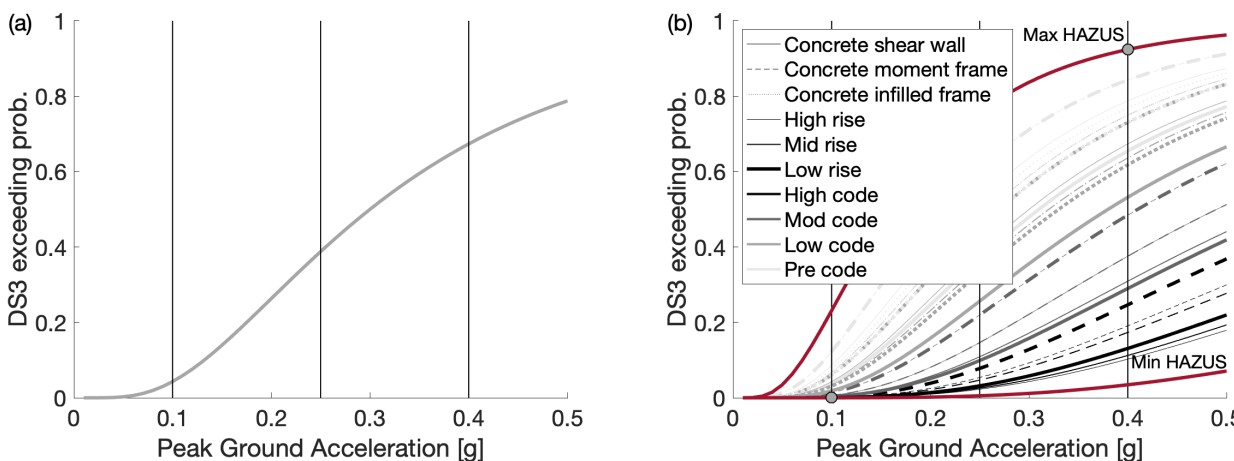

**Figure 2 a) Example baseline score for a given archetype building typology. b) HAZUS fragility curve database related to the Extensive Damage limit state for RC buildings.**

$$I_{BL} = \left( \frac{50 - 1}{P_{HAZUS,max} - P_{HAZUS,min}} \right) (P_{HAZUS} - P_{HAZUS,min}) + 1 \qquad\qquad 3$$

### 3.2. Performance modifier

Eight secondary parameters are used to define the performance modifier, which are deemed not explicitly considered in the HAZUS framework (and therefore in the baseline score) but at the same time, if present, can jeopardise the seismic performance of a given building. Firstly, based on Table 3, a score is assigned for each secondary parameter ($SCORE_i$ in the range [0%, 100%]). Therefore, the performance modifier ($\Delta I_{PM}$) is defined as a weighted sum of each of these scores (Eq. 4, where $w_i$ is the weight of parameter $i$), finally scaling the result in the range [0%, 50%]. It is worth mentioning that, according to this definition, the performance modifier can only increase the baseline score, therefore denoting an increase in seismic fragility. For a more simplified utilisation, it is also possible to calculate the INSPIRE index considering the baseline score

only. In such case, a default performance modifier equal to 25% is assigned (average of 0% and 50%). It is worth mentioning, however, that any uniform value of the performance modifier will not have effects on the overall prioritisation, since it will shift all the calculated indices by the same amount.

$$\Delta I_{PM} = \frac{1}{2} \sum_{i=1}^{8} w_i SCORE_i \qquad\qquad 4$$

**Table 3 Secondary parameters: definition, available alternatives, related scores and weights.**

| Secondary parameter | Scores | Alternatives | Weight |
|---|---|---|---|
| Preservation condition and/or existing damage | 100 | Significantly affecting performance | 0.0939 |
| | 50 | Moderately affecting performance | |
| | 0 | Not affecting performance | |
| Plan shape | 100 | L-shape or irregular | 0.0826 |
| | 50 | C-shape | |
| | 0 | Rectangular or regular | |
| Storey height uniformity | 100 | Significantly non-uniform (more than 0.5m difference) | 0.0470 |
| | 50 | Moderately non-uniform (difference between 0 and 0.5m) | |
| | 0 | Uniform | |
| Added Storeys | 100 | Yes | 0.0470 |
| | 0 | No | |
| Infills at ground storey | 100 | No | 0.3039 |
| | 0 | Yes | |
| Short column | 100 | Yes | 0.1817 |
| | 0 | No | |
| Pounding | 100 | Pronounced (less than 0.1m gap) | 0.1817 |
| | 50 | Moderate (gap between 0.1m and 0.2m) | |
| | 0 | None (more than 0.2m gap) | |
| Unfavourable soil | 100 | Yes (very soft soil. Liquefaction is not explicitly | 0.0621 |
| | 0 | considered) | |
| | | No | |

The secondary parameters defining the performance modifiers are deemed to complement the information in the HAZUS fragility curves. The idea is that the baseline score (HAZUS fragility database) provides the (conditional) seismic risk of a given building class, while the secondary parameters are related to building-specific vulnerability factors. The secondary parameters have been selected based on the fundamental rules of seismic design (e.g., Paulay and Priestley, 1992) and the commonly observed post-earthquake damage on RC structures (e.g., Elnashai and Di Sarno, 2008, Palermo et al., 2017, De Luca et al., 2018). For each of them, Table 3 provides guidance on the selection of the alternatives. The score for each alternative has been defined based on a uniform partitioning of the range [0%, 100%]**.**

Evidently, the secondary parameters defining the performance modifier do not have the same influence on the overall vulnerability and risk. For example, the absence of infill walls at the ground storey can lead to a soft-storey mechanism, which in turn can results in local or global collapse. On the other hand, the addition of storeys to a building can increase its risk to a

lower extent, considerably less likely leading to collapse. Therefore, the weight of the former parameter should be higher than the latter, to reflect such a different effect on the overall seismic risk.

To this extent, the Analytic Hierarchy Process (AHP), originally proposed by Saaty (1980) is used to calibrate the weights in the proposed procedure. This allows to have a rational and mathematically-consistent assignment of the weights which is based on pairwise comparisons between the secondary parameters and eigenvalues theory. Hence, the selected weights can reflect the relative importance of each parameter with respect to the others in the determination of the performance modifier. It is worth mentioning that such approach has been successfully adopted in other earthquake engineering applications, such as the selection of the optimal seismic retrofitting strategy for case-study buildings (Caterino et al., 2008). Also, it is worth mentioning that, in its current version, the expert judgement defining the weights adopted in the procedure is provided by the authors. However, such weights can be further updated considering the opinion of a group of experts in the field of structural and earthquake engineering.

After the definition of the parameters involved in the analysis, the first step of the process consists of expressing an expert opinion about every possible pairwise comparison of those parameters. Given two parameters $P_i$ and $P_j$, the relative importance of $P_i$ over $P_j$ is expressed with the coefficients $a_{ij}$, defined according to Table 4. For the calibration proposed in this paper, the pairwise comparisons are performed considering the relative influence of the secondary parameters on the Life Safety performance objective. The related judgement matrix [**A**] containing the 27 pairwise comparisons is given in Table 5. As an example, the presence of infill walls at the ground storey has been considered extremely more important than the presence of unfavourable soil, given its implications on Life Safety. Therefore, the related $a_{ij}$ parameter is set to 9.

**Table 4 Scale of relative importance of the secondary parameters (Saaty, 1980).**

| Relative importance ($a_{ij}$) | Description |
|---|---|
| 1 | Parameters $P_i$ and $P_j$ are equally important |
| 3 | Parameter $P_i$ is moderately more important than $P_j$ |
| 5 | Parameter $P_i$ is essentially more important than $P_j$ |
| 7 | Parameter $P_i$ is demonstratedly more important than $P_j$ |
| 9 | Parameter $P_i$ is extremely more important than $P_j$ |
| 2, 4, 6, 8 | Intermediate values between the two adjacent judgements |
| Reciprocal of the above | If criterion i compared to j gives one of the above, then j, when compared to i, gives its reciprocal |

Once the pairwise comparisons have been performed, the vector {**w**} containing the weights of the secondary parameters is found by solving the eigenvalue problem $Aw = \lambda_{max}w$, where $\lambda_{max}$ is the largest eigenvalue. The principal right eigenvector of the [**A**] matrix is the vector of the weights {**w**}, after normalisation with respect to its sum. Using such approach allows to measure the consistency of the pairwise comparisons, therefore minimising the chance to have a random prioritisation of the parameters. In fact, if the pairwise comparisons are performed in a perfectly consistent manner, the [**A**] matrix has only one eigenvalue equal to its rank, and the coefficients $a_{ij}$ are equal to the ratio $w_i/w_j$ of the weights related to the parameters $P_i$

and $P_j$. Practically, the comparisons are unlikely to be perfectly consistent, and the first eigenvalue of the [**A**] matrix will be slightly different than its rank, while the other eigenvalues are close to zero.

**Table 5 Judgement matrix adopted for the calibration of the weights.**

|  | Preservation condition | Plan shape | Storey height uniformity | Added Storeys | Infills at ground storey | Short column | Pounding | Unfavourable soil |
|---|---|---|---|---|---|---|---|---|
| Preservation condition | **1** | 1 | 2 | 2 | 1/3 | 1/2 | 1/2 | 2 |
| Plan shape | 1 | **1** | 2 | 2 | 1/3 | 1/2 | 1/2 | 1/2 |
| Storey height uniformity | 1/2 | 1/2 | **1** | 1 | 1/6 | 1/4 | 1/4 | 1 |
| Added Storeys | 1/2 | 1/2 | 1 | **1** | 1/6 | 1/4 | 1/4 | 1 |
| Infills at ground storey | 3 | 3 | 6 | 6 | **1** | 2 | 2 | 6 |
| Short column | 2 | 2 | 4 | 4 | 1/2 | **1** | 1 | 4 |
| Pounding | 2 | 2 | 4 | 4 | 1/2 | 1 | **1** | 4 |
| Unfavourable soil | 1/2 | 2 | 1 | 1 | 1/6 | 1/4 | 1/4 | **1** |

Therefore, the consistency of the pairwise comparison is measured by calculating the Consistency Index (CI) using Eq. 5, where $n$ is the rank of the matrix. The CI is compared to the Random Consistency Index (RCI), which is the average Consistency Index of a large number of randomly-generated reciprocal matrices using the scale (1/9, …, 1, …, 9). According to Saaty, 1980, for 8x8 matrices the Random Consistency Index is equal to 1.41. According to the same study, if the CI is

10 smaller than 10% of the RCI (Eq. 6), the final choice of the weights is logically sound and not a result of random prioritisation. In general, if such criterion is not satisfied for the performed pairwise comparisons, the whole process should be repeated until acceptable consistency is achieved. For the pairwise comparisons in this paper, the consistency index is equal to 2.1%, which is considered acceptable.

$$CI = \frac{\lambda_{max} - n}{n - 1}$$

$$\frac{CI}{RCI} \leq 0.1$$

**3.3. Multi-hazard considerations**

15 In many common situations, it is very likely that seismic considerations alone are not enough to define a robust prioritisation scheme for the decision-making process by a governmental agency or the owner of a large building portfolio. To deal with such scenario, this paper presents a simple methodology to include other hazards in the prioritisation scheme.

Simplified analytical indices are available for the estimation of the vulnerability of buildings to natural hazards different than seismic hazard, such as Tsunami (e.g., Dall'Osso et al., 2016), Flood (e.g., Stephenson and D'Ayala, 2014, Pazzi et al., 2016, Nassipour et al., 2018), Wind (e.g., Womble et al., 2016), etc. Once the desired single-hazard vulnerability indices ($I_k$) are computed, a combined, multi-hazard index can be defined as the multi-dimensional distance from the origin of the system of reference (Eq. 7). Although this concept is applicable regardless of the number of considered hazards, a two-dimensional example is presented in Figure 3. Supposing that both "hazard 1 and 2" are defined over a range [1%, 100%], the combined index will be defined on the range [1%, 141.4%]. However, this can be re-scaled in any other desired range without affecting the prioritisation list of the considered building portfolio. It is worth mentioning that cumulative damage related to subsequent hazards are outside the scope of such a simplified multi-hazard index.

$$I_{multi} = \sqrt{\sum_k I_k^2}$$

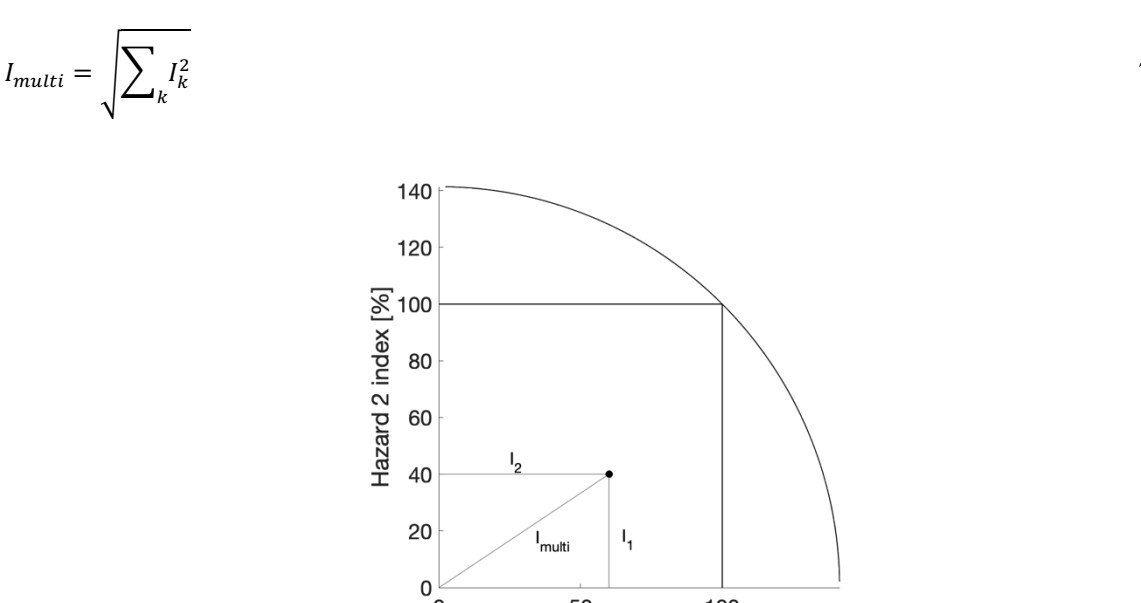

**Figure 3 Definition of a two-dimensional ($k = 2$) multi-hazard index.**

## 3.4. Dealing with subjectivity

According to the definition of the seismic risk prioritisation index given in *Section 3*, it appears evident that a degree of subjectivity is always involved in the calculation. In particular:

1) The baseline score is based on the fragility functions available in the HAZUS MH4 guidelines. Although such model is largely based on numerical analyses of building models, the results are provided for a limited number of structural categories. The user should carefully select the category that best matches with the characteristics of the considered buildings, with special reference to the "Code level" parameter;

2) The weights needed for the calculation of the performance modifier are based on a subjective set of pairwise comparisons between the secondary parameters, although this is derived in a mathematically-consistent and rational way that allows to minimise the chance of having randomly-assigned weights;

3) The ratio between the maximum possible baseline score and the maximum possible performance modifier is "arbitrarily" set to unity, i.e. $I_{BL,max} = 50\%$ and $\Delta I_{PM,max} = 50\%$.

It is worth noting that the subjectivity is an intrinsic component of any simplified vulnerability index, including the INSPIRE index. Therefore, rather than assessing a single building, these methods should only be used to derive a proxy for the relative vulnerability (or risk) of buildings in a given portfolio, and to define a prioritisation scheme for possible risk mitigation actions. However, in this section some measures are proposed to control and minimise the subjectivity involved with the definition of the index. An example of such measures is given in *Section 4*, in relation to a real building portfolio application.

It is virtually impossible to perfectly match the considered buildings with the archetype buildings in HAZUS. However, a careful examination of the characteristics of the considered buildings should be carried out, to better select the appropriate HAZUS categories. Characteristics such as the presence of strong beams vs weak columns should be considered, which can lead to a Pre-Code classification, or documented structural retrofit measures, which can lead to higher "Code level" classification. A review of the (history of the) structural/seismic codes appropriate for the considered buildings can considerably reduce the level of subjectivity. These can be compared to the different provisions in UBC1994, defining "equivalence relationships". Any information related to the year of construction (or design) of the considered buildings is fundamental in such process. Overall, it is deemed that the assignment of the HAZUS categories to the considered buildings should reflect their expected differences in their seismic performance, rather than perfectly match the properties of the archetype in the HAZUS category definition.

As an alternative to the HAZUS definition, the baseline score can be re-defined adopting, if available, a portfolio-specific set of fragility curves. To this aim, the OpenQuake platform (open-source, https://storage.globalquakemodel.org/openquake), by the Global Earthquake Model (GEM) foundation, might be used. Among other capabilities, such platform contains large databases of empirical and numerical fragility/vulnerability models appropriate for many structural typologies and many regions of the world.

An illustrative set of weights needed to calculate the performance modifier is given in this paper. However, a case-specific AHP, for instance involving groups of local experts, can be performed to derive new weights that match the characteristics of the considered building portfolio. Such a procedure cannot remove the subjectivity related to the performance modifier but provides the user with a tool to reduce it and have a close match between the analysed building portfolio and the adopted weights.

Finally, the subjectivity related to the ratio between the maximum possible baseline score and performance modifier can be tested through a sensitivity analysis. For a given building portfolio, the idea is to repeat the calculation of the seismic index with different values of the maximum possible performance modifier. This allows to check the reliability of the priority list to

this assumption. If slight modifications in the maximum possible performance modifier lead to high differences in the resulting priority list, engineering judgement should be adopted to justify the final choice.

## 4. Illustrative application: school building portfolio in Banda Aceh, Indonesia

### 4.1. Description of the case-study portfolio and definition of an archetype building

The case study portfolio selected for this study consists of 85 RC buildings belonging to 44 school compounds located in Banda Aceh, the capital and largest city in the province of Aceh, Indonesia (Figure 4). Banda Aceh is located in the island of Sumatra and, according to the 2000 census, has a population of 219,070 people (Seta, 2000). The city was severely affected by the 26 December 2004 Indian Ocean earthquake (Moment magnitude, $M_w$=9.1), being the closest major city to the event location. Due to the effects of the following devastating tsunami, the city suffered from 167,000 deaths and catastrophic

damage to structures and infrastructures. It is worth noting that Indonesia suffered a 4.5bln\$ economic loss for this event (Pomonis et al., 2006). In the Aceh region, 45,000 students and 1,870 teachers died and 2,065 education facilities were damaged, 100,000 (BAPPENAS, 2006).

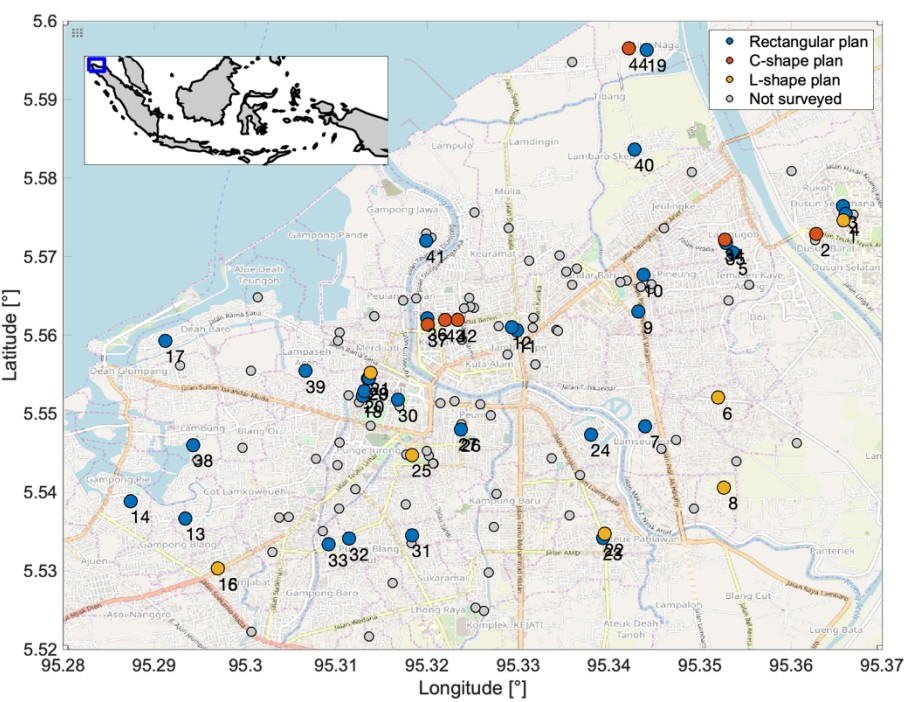

**Figure 4 Map of the school buildings in Banda Aceh, including non-surveyed ones.**

A RVS campaign through the INSPIRE form (Figure 1) was carried out to collect administrative, geometric and mechanical data related to the investigated buildings. The RVS campaign (Figure 5) was conducted by three teams composed of one experienced engineer and two final-year (one undergraduate and one postgraduate) local engineering students working for four

days. The surveys were conducted in all the suburbs of Banda Aceh (Figure 6a), to obtain a clear overview of the construction practice in the city. For all the surveyed buildings, the BSS is a reinforced concrete frame with infills. The majority (81%) of the buildings in the portfolio has a rectangular plan, with the remaining 19% composed of L-, C- or T-shaped plans (Figure 6b). 68% of the surveyed buildings is two-storey high, while one- and three-storey buildings represent 15% and 16% of the portfolio, respectively (Figure 6c). The year of construction for each building was retrieved from the school signboard, registry or by interview of the school principal. Figure 6d shows that few buildings (18%) survived the 2004 earthquake-tsunami sequence; hence, the majority of the portfolio (57%) was constructed between 2005 to 2011, while the remaining 25% was built from 2013 onwards.

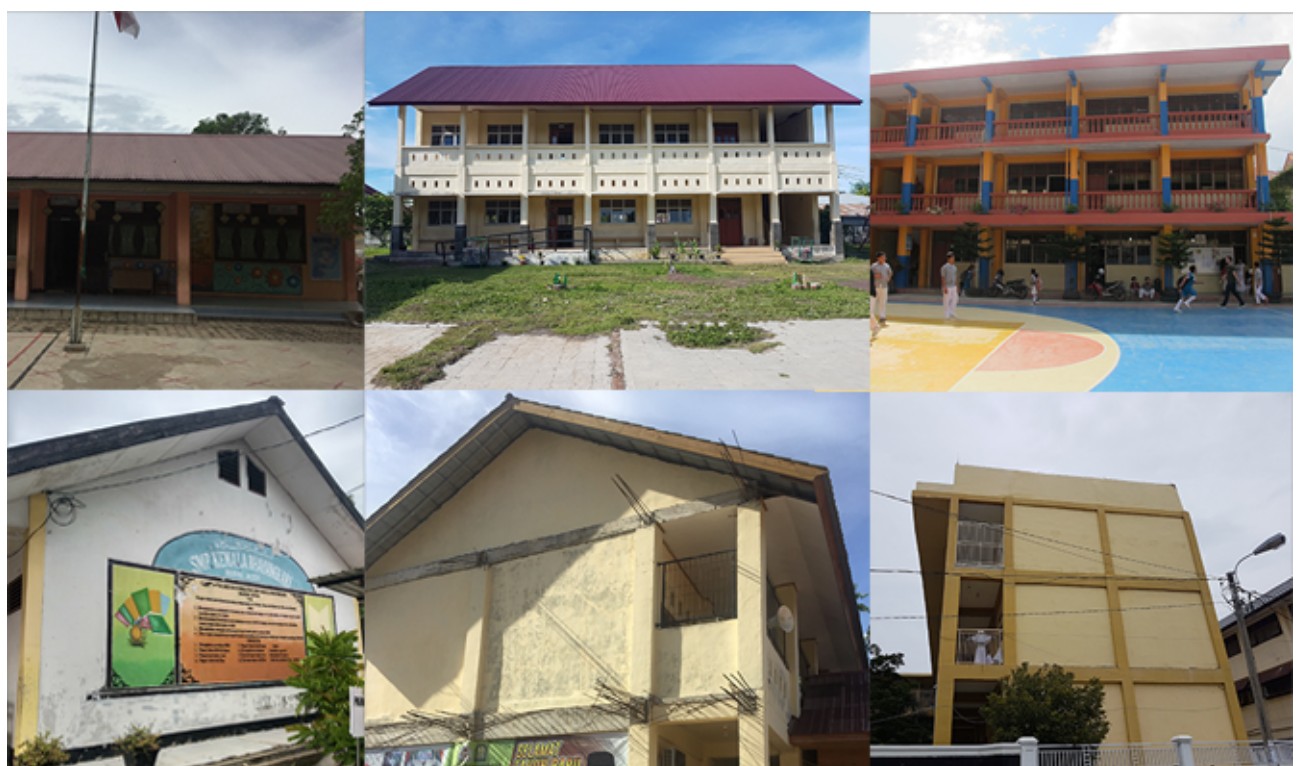

**Figure 5 Sample photos of the building portfolio (taken on 16-19 October 2018).**

The accurate knowledge of the year of construction is essential for the structural characterisation of the portfolio. In fact, such information can be coupled with the history/evolution of the available structural and/or seismic codes to derive minimum-by-law dimensions of the structural members, reinforcement detailing, level of considered vertical and lateral forces in the design, etc. In this specific case, the appropriate structural code for the whole portfolio is the SKBI 1.3.53.1987 (SNI, 1987). The first seismic code appropriate for the region is the SNI 1726:2002 (SNI, 2002), which is inspired to the American Uniform Building Code 1997 (UBC 1997), which also facilitates the compatibility with the HAZUS framework. Stricter seismic provisions were adopted when the seismic code was updated in the SNI 1726:2012 (SNI, 2012), which fully consistent with the American

ASCE 7-10 (ASCE, 2010). Therefore, apart from a minority (12) of buildings constructed before 2002, approximately half of the buildings are constructed according to the SNI 1726:2002 (lower) standards while the other half refers to the updated SNI 1726:2012, with nominally better nominal seismic performance.

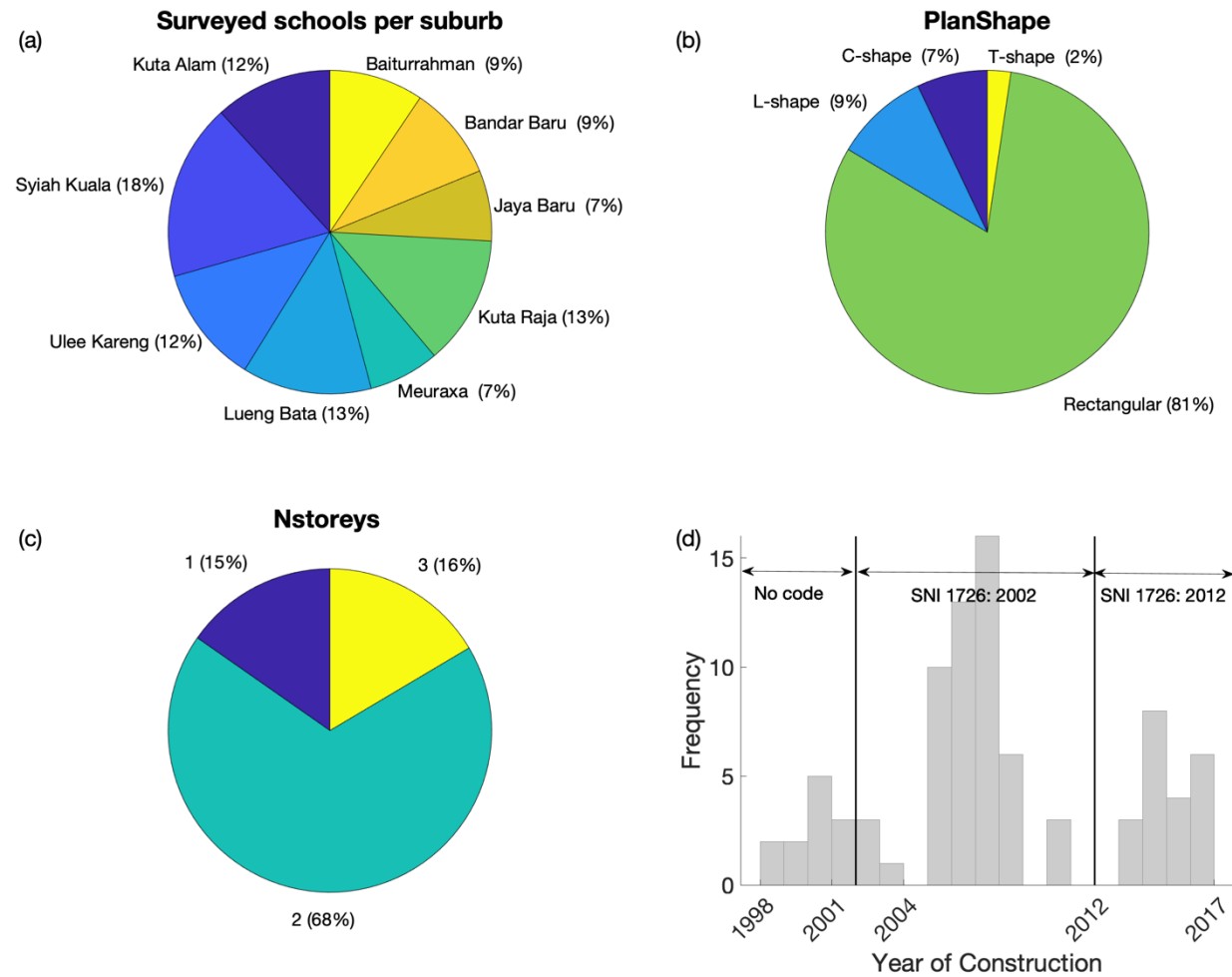

5    **Figure 6 Statistics for the 85 surveyed school buildings.**

The analysis above, based only on the information of the INSPIRE forms, allows to identify an archetype building which represents the construction practice for school building in Banda Aceh. The archetype building is a two-storey, rectangular RC

building.

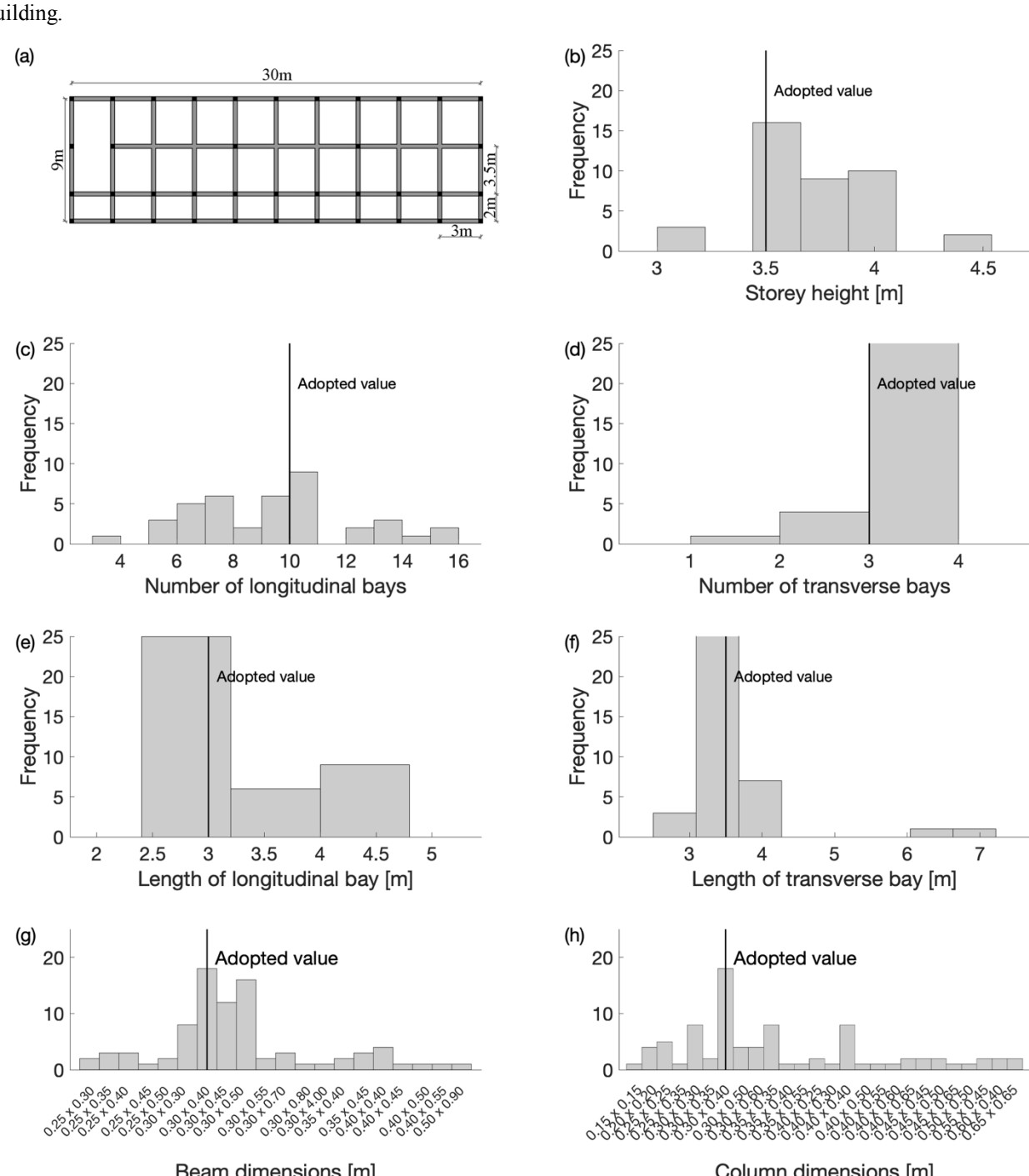

Figure 7 shows the geometrical characteristics of the archetype building geometry, which are based on the modal (most frequent) values of the empirical distributions (histograms) considering only the rectangular, 2-storey buildings in the portfolio

(69). The archetype has 10 bays in the longitudinal direction (one for the staircase, three for each classroom). In the transverse direction there are two 3.5m bays and a 2m corridor bay. The storey height is equal to 3.5m. 0.4x0.3m columns are adopted, except for the corridor columns, whose dimensions are 0.3x0.3m. Finally, the dimensions of the typical beams are 0.3x0.4m. The dimensions of beams and columns are validated with simulated design approaches according to the above-mentioned codes.

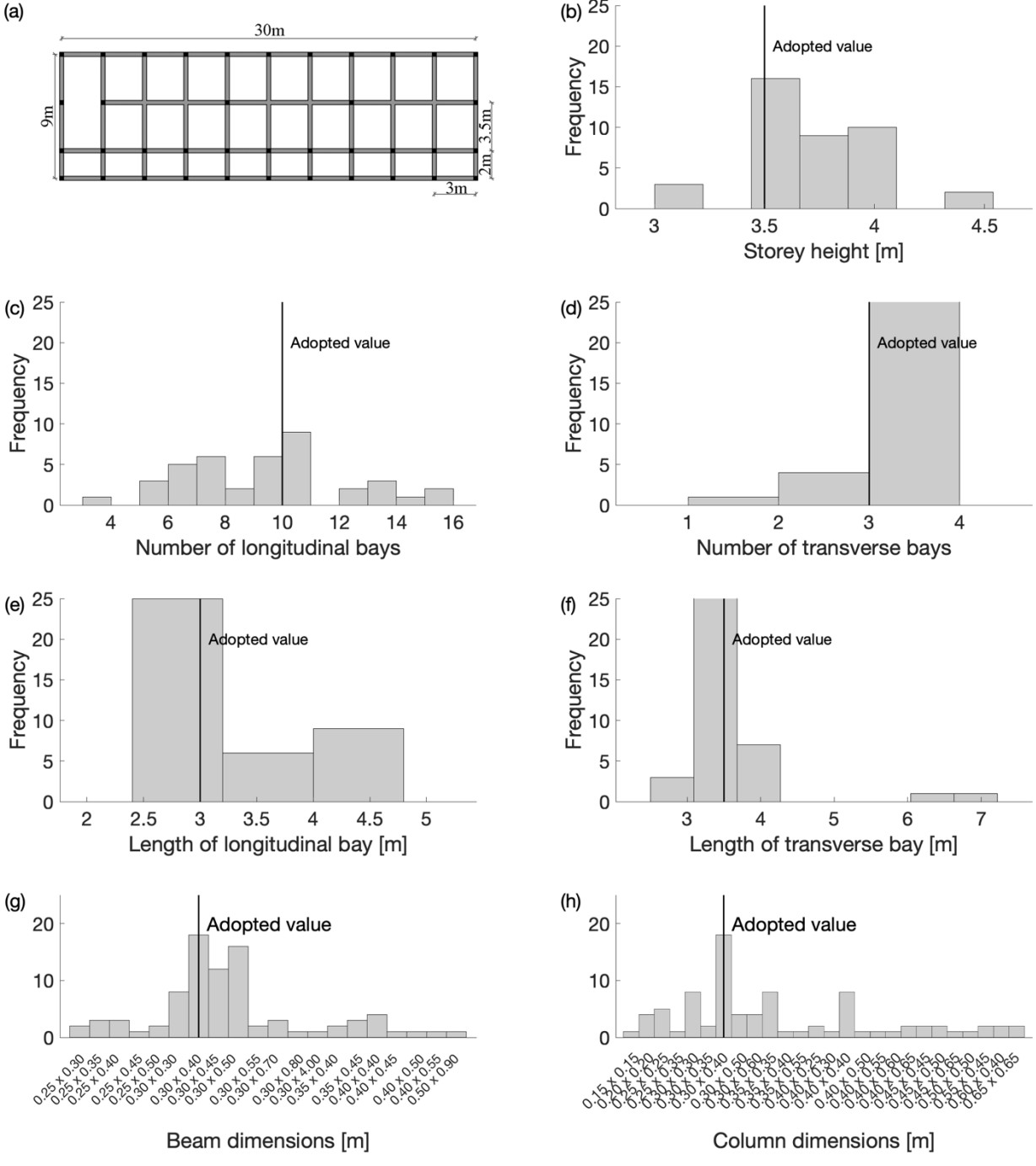

**Figure 7 Two-storey, rectangular buildings: geometry trends and adopted values for the archetype building. a) plan view of the archetype building; b-h) histograms of the geometric parameters.**

Considering the same geometry, two different sets of structural details are provided, to reflect the SNI2002 and SNI2012 seismic codes. Table 6 provides examples of structural reinforcement for typical members. It is worth mentioning that, for the Post-2012 archetype structure, the cross-section height of beams and columns are respectively 5cm and 10cm bigger than the corresponding members in the Pre-2012 archetype structure. The reinforcement of the structural members is selected by cross-

checking visual information (see Figure 5) with the outcome of both gravity-based and seismic-based simulated design according to the SNI codes. To this aim, acting loads are calculated considering permanent dead loads (according to the suggested material properties in SNI1987) and live load equal to 5kPa (1kPa for the roof). Gravity axial load on columns is calculated based on a tributary area approach. This cross-checking exercise has shown that the real observed amount of longitudinal reinforcement is greater than the minimum by code. On the other hand, based on the limited visual information

available for the transverse reinforcement, no joint stirrups are conservatively considered for both the Pre-2012 and Post-2012 classes, regardless of the requirement in both codes.

**Table 6 Typical structural details for the archetype building(s).**

|  | Materials (mean values) | Typical beam | Typical column | Typical joint |
|---|---|---|---|---|
| Pre-2012 | Concrete $f_c$ = 21.5MPa | $3\phi16$ top | $3\phi16$ top | No stirrups |
|  | Long. Steel $f_y$=400MPa | $3\phi16$ bottom | $3\phi16$ bottom |  |
|  | Tran. Steel $f_y$=240MPa | $\phi10$@150mm stirrups | $\phi10$@200mm stirrups |  |
| Post-2012 | Concrete $f_c$ = 24MPa | $3\phi16$ top | $3\phi16$ top | No stirrups |
|  | Long. Steel $f_y$=400MPa | $3\phi16$ bottom | $3\phi16$ bottom |  |
|  | Tran. Steel $f_y$=240MPa | $\phi10$@150mm stirrups | $\phi10$@100mm stirrups |  |

Note: $f_c$ is the concrete compressive cylinder strength; $f_y$ is the steel yield stress.

## 4.2. Prioritisation scheme

Based on the data collected with the forms, the INSPIRE seismic risk prioritisation index is calculated for the whole portfolio. Moreover, the Tsunami PTVA4 index (Dall'Osso et al., 2016) is calculated, finally combining these results to derive a multi-hazard index. It is worth mentioning that the resulting indices values are arbitrarily categorised in three groups, respectively "green, yellow and red tags" by defining two threshold values for the various indices. The definition of such thresholds is essentially a subjective (often political) choice that shapes the prioritisation scheme, based for instance on resources

availability. For a governmental agency, those can be calibrated estimating the average structural retrofit (or relocation) cost per building and defining the amount of available public funding in two or more-time windows (e.g. one and five years) to obtain specified risk-reduction objectives. As a proof of concept, in this paper the thresholds are selected to be equal to 33% and 66% for the calculated seismic, tsunami or multi-hazard indices.

The PTVA index, similarly to the proposed INSPIRE index, allows to derive the relative tsunami vulnerability of a building.

It is calculated as a weighted sum of two parameters: the "structural vulnerability" and the so-called "water vulnerability". The first parameter depends on three factors: the type of the lateral resisting system, the depth of the flood water at the building location, and the degree of external protection (e.g. presence of seawalls). The "water vulnerability" depends on the ratio of the inundated-to-total number of storeys. It is worth mentioning that this parameter is calculated using the inundation data

from the 2004 Indian Ocean Tsunami. In particular, Iemura et al., 2012 provide 85 field-measurements of the tsunami height (from the ground) for the city of Banda Aceh. For the purpose of this paper, a linear regression ($R^2$=0.66) is performed to define a linear relationship between distance from the coast and tsunami height. The calculation of the final risk-related index is dependent on scoring parameters assigned according the number of storeys of the building, its material, the percentage of openings (e.g. windows), foundation type, impacting objects, orientation and shape of the building, and its position with respect to a building row. It is worth mentioning that, although the name refers to vulnerability, the PTVA index somehow refers to tsunami risk, since the expected hazard is also considered. This facilitates the compatibility with the INSPIRE index.

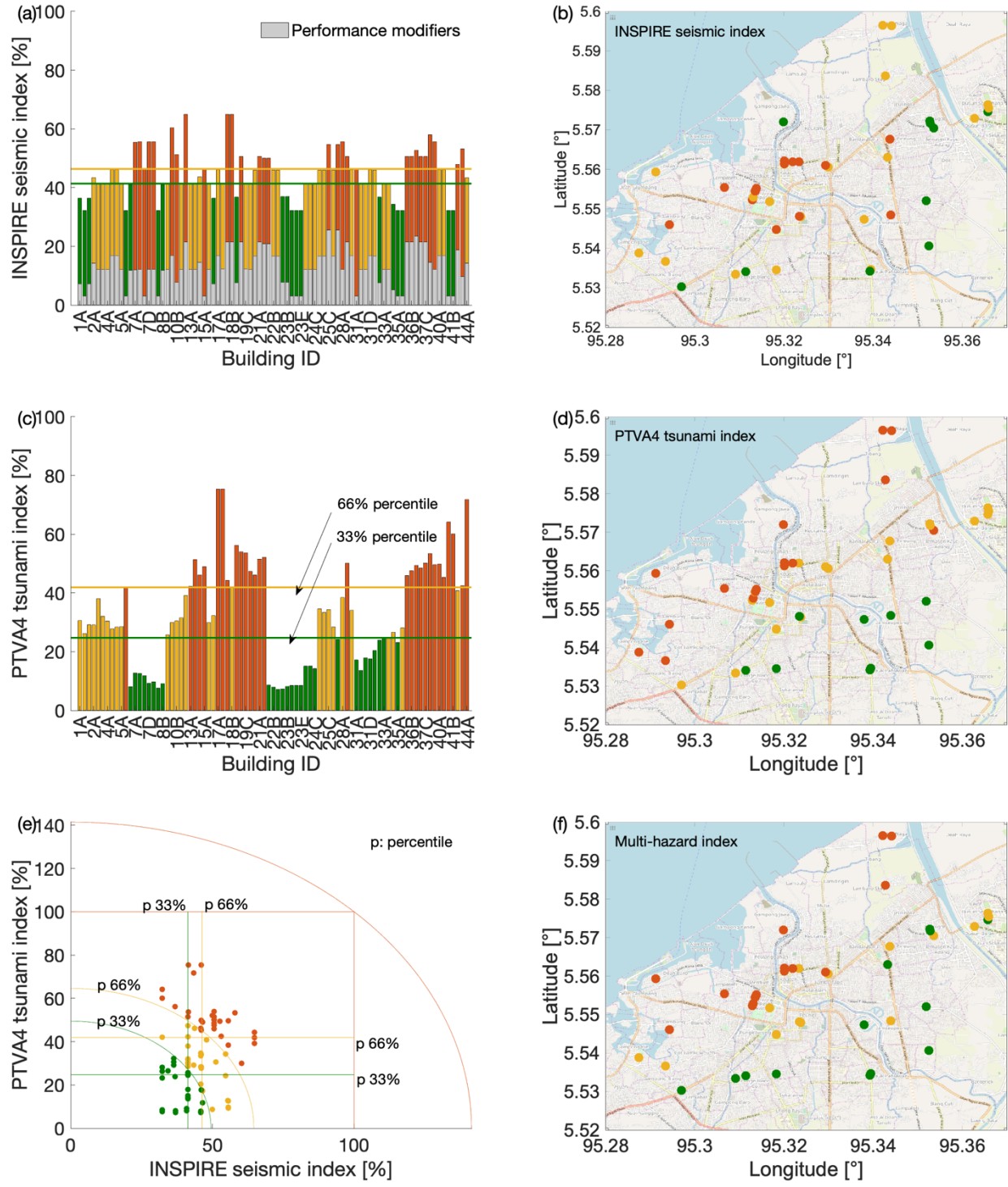

**Figure 8 Application to the seismic and tsunami indices to 85 RC school buildings in Banda Aceh, Indonesia.**

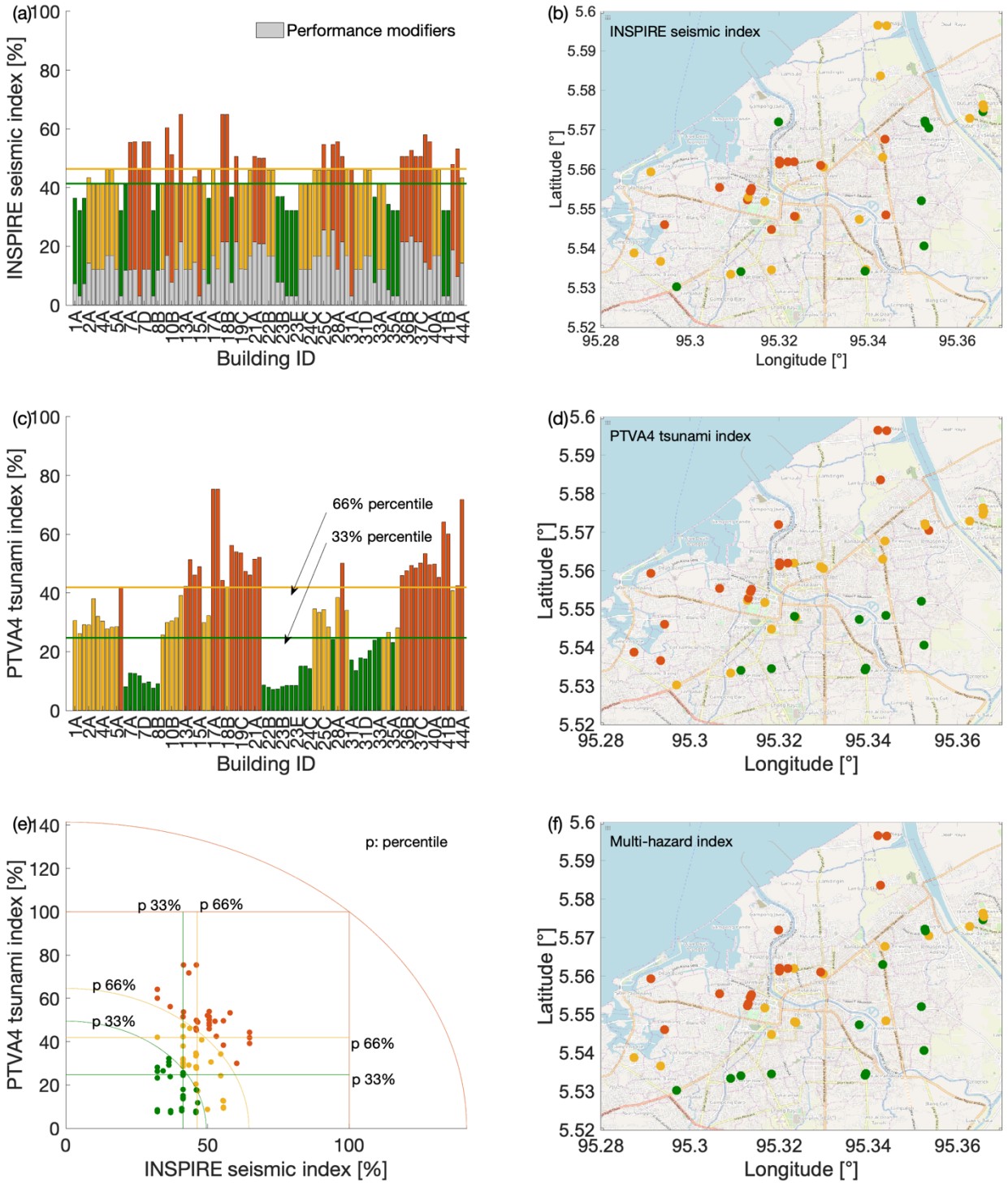

Figure 8a,b show the application of the INSPIRE seismic risk prioritisation index to the considered portfolio. To calculate the baseline scores, the HAZUS fragility curves related to the C1 category "Concrete Moment Frame" are used. Since the infills

of the investigated frames are made of a single layer of poor-quality clay bricks, their presence is neglected. According to the analysis of the year of construction and the history of the structural/seismic codes in Indonesia, the categories Pre-Code and Low-Code are adopted for the Pre-2012 and Post-2012 buildings, respectively. Given the particularly small differences in the characteristics of the buildings, the INSPIRE index is particularly similar for the whole portfolio [32%, 64%]. The slight differences in the value of the index are due to the performance modifiers, mainly governed by short columns and/or pounding for the majority of the schools. This is a further confirmation of the uniformity of the construction practice for school buildings in Banda Aceh, also observed in other countries (e.g., Nassirpour et al., 2018, Zhou et al., 2018). It is worth mentioning that, for this particular portfolio, soil condition is not influencing the performance modifier, since the "unfavourable soil" parameter

is set to "yes" for all the buildings. On the other hand, the results for the PTVA tsunami index (

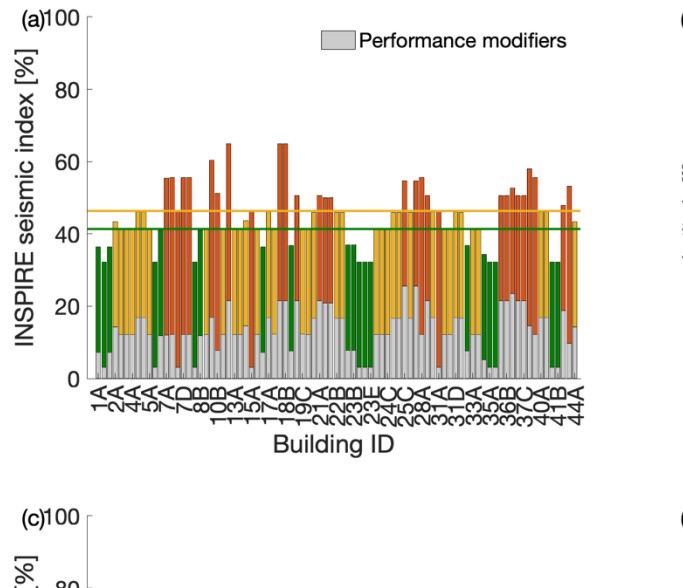

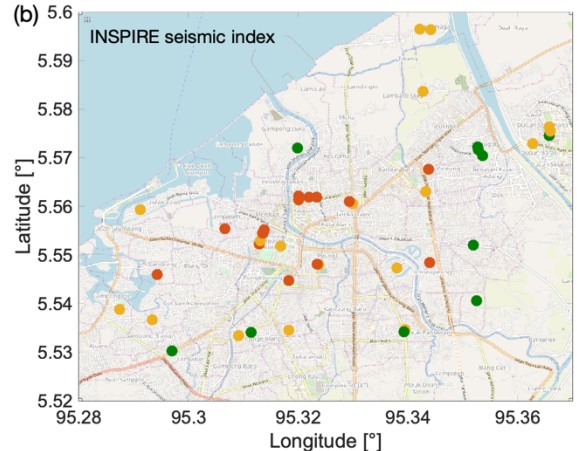

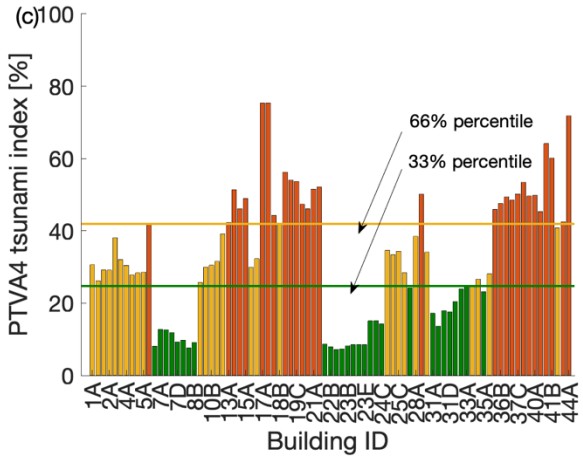

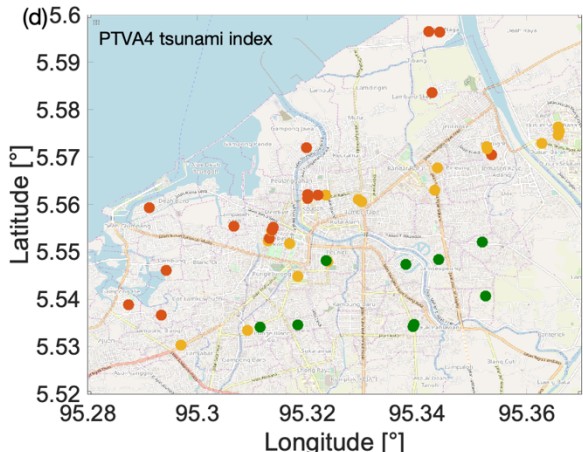

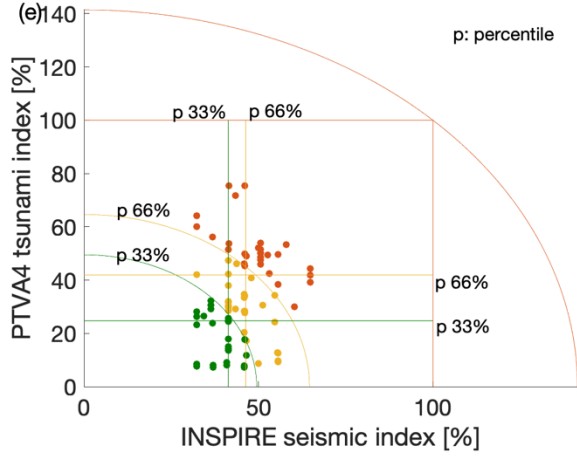

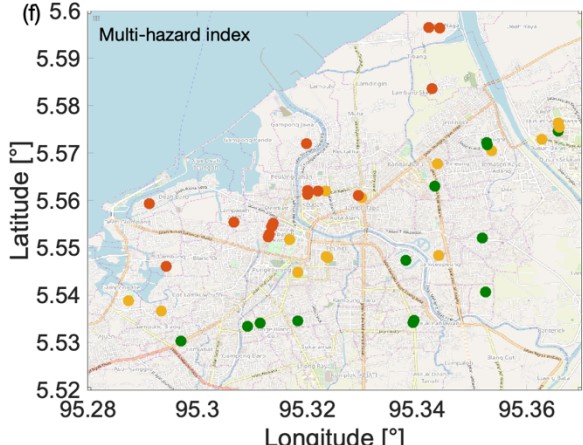

Figure 8c,d) show a larger variability [7%, 75%] and a clear correlation between the distance from the coast and the relative tsunami risk. This result, although preliminary, might suggest that the distance from the coast can be used as a very simple proxy for the "water vulnerability" parameter in the tsunami PTVA index.

Given the above-mentioned trends, the combination of the seismic and tsunami indices (

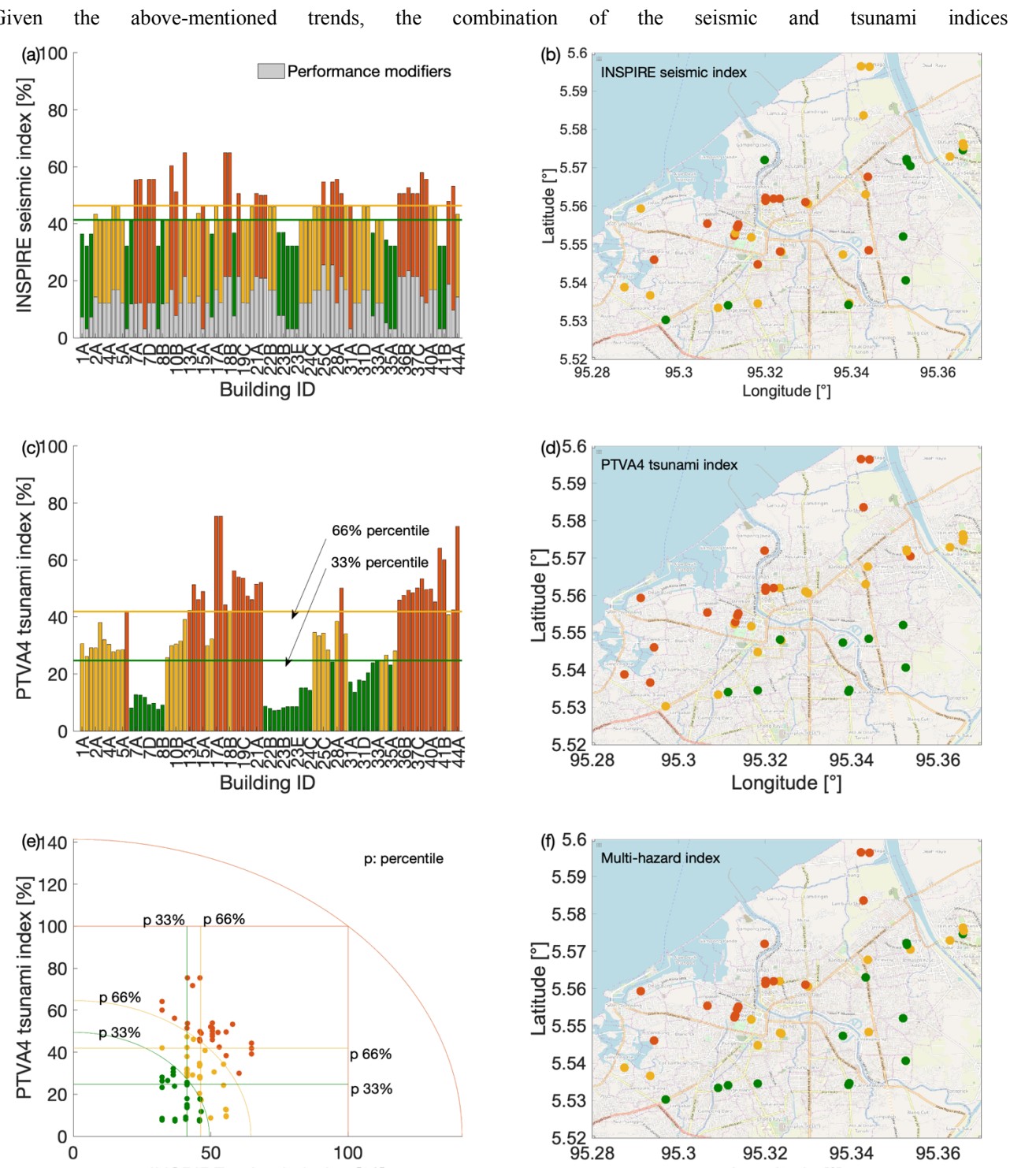

Figure 8e,f) clearly indicates that the tsunami hazard play a substantial role in determining the prioritisation scheme for the school buildings in the city of Banda Aceh. Indeed, the overall trend of the multi-hazard results is practically equal to the trend of the tsunami index results. For instance, the developed maps could be used to identify "safer areas" where strategic buildings (e.g., schools or hospitals) should be located, increasing the awareness of vulnerabilities that could be integrated in emergency planning for critical infrastructure disruption (e.g., Pescaroli and Alexander 2016).

To control the role of the subjectivity in calculating the INSPIRE seismic index, a sensitivity study is conducted. The seismic index is applied to the entire portfolio four times (

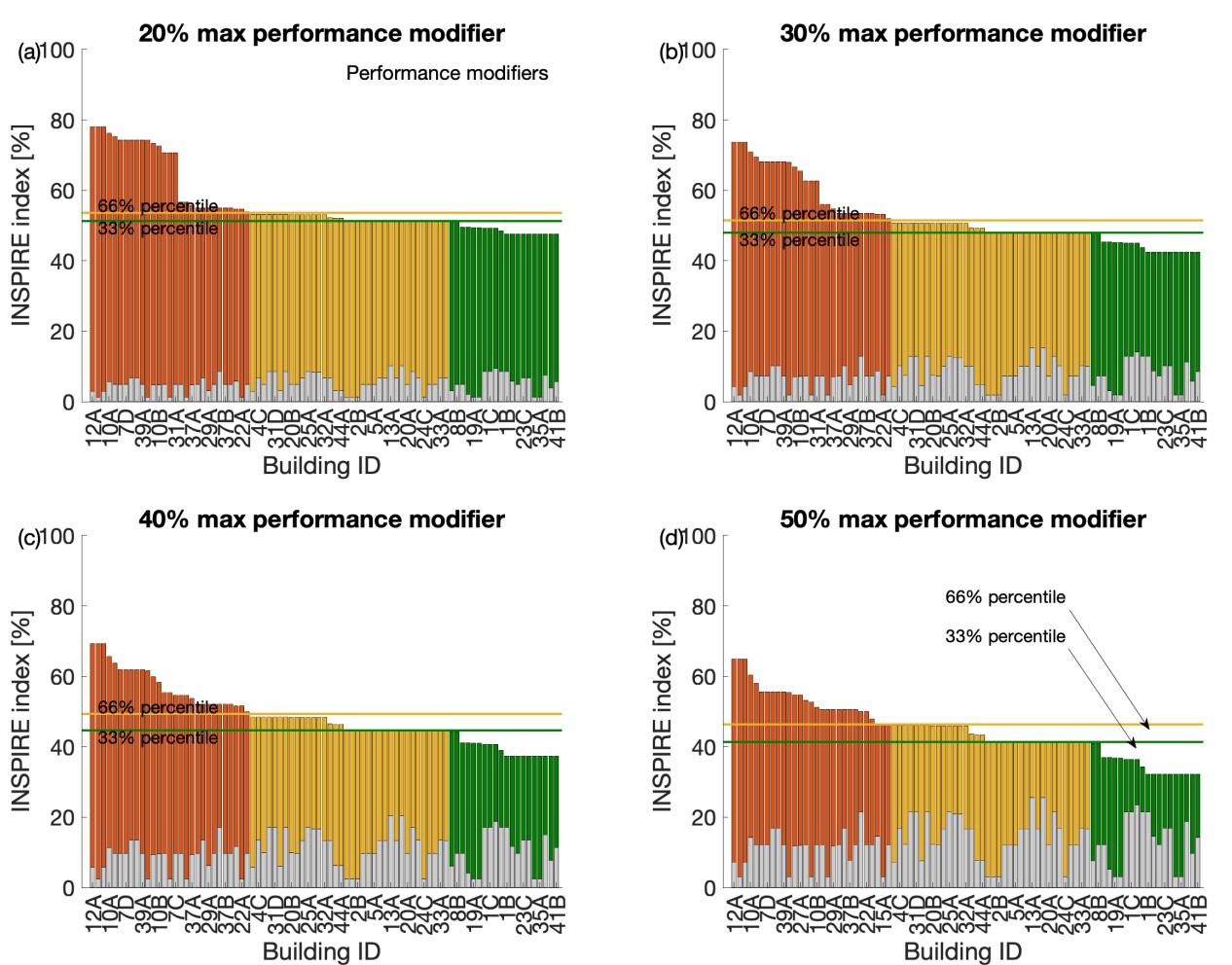

Figure 9), by considering that the maximum value of the performance modifier is equal to 20%, 30%, 40%, and 50%. The maximum value of the baseline score is respectively equal to 80%, 70%, 60%, and 50%.

The results of the seismic index are ranked in descending order of risk. The results clearly show that, for this portfolio, the overall priority list is rather insensitive to the maximum baseline-to-performance modifier ratio. Indeed, a small number of

"swaps" in the priority list is sought and, with the same definition of the thresholds for the tags, the number of building in each category has a negligible dependency on the ratio above.

### 4.3. Analytical/numerical seismic fragility analyses for the archetype building(s)

As discussed above, the INSPIRE form provides enough data to build refined numerical models for one or more selected buildings, in a multi-level approach. While the previous sections illustrate the portfolio-level approach (relative-risk prioritisation), this section illustrates a possible structure-specific application of the proposed framework. To this aim, the archetype building(s) defined in *Section 4.1* are further analysed by means of non-linear static and non-linear dynamic analyses to derive structure-specific fragility curves. Those are derived considering a two-dimensional representation of the longitudinal and transverse frames that compose the archetype building. As discussed above, due to their small thickness and poor quality, infills are not considered in the models. Both the Pre-2012 and the Post-2012 archetype buildings are analysed, leading to four different computational models.

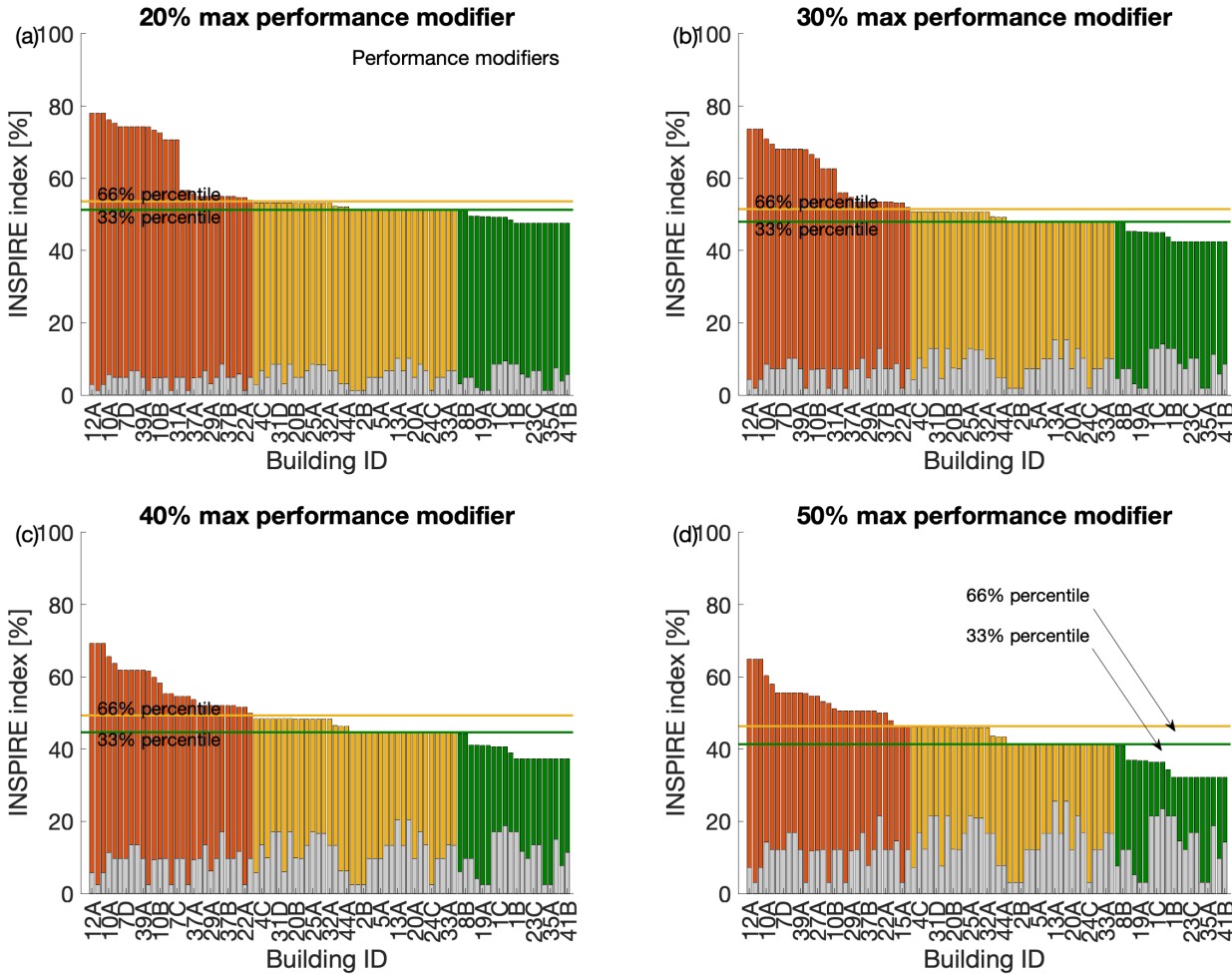

**Figure 9 Sensitivity analysis for the baseline-to-performance modifier ratio.**

The Simple Lateral Mechanism Analysis (SLaMA) is firstly used to obtain a first estimation of the non-linear force-displacement curve of the building. SLaMA (NZSEE 2017, Gentile et al., 2019a,b,c,d) is a tool to derive both the expected plastic mechanism and the capacity curve of RC frame, wall and dual-system buildings by using a "by-hand" procedure (i.e., using an electronic spreadsheet). This allows to identify potential structural weaknesses in the lateral resisting mechanism and allows to test the reliability of numerical computer models in capturing the most probable behaviour of a structure. It is worth mentioning that, each beam and column in the system is characterised considering many possible failure mechanisms (i.e., flexure, bar buckling, lap-splice failure, shear), considering that the weakest link will govern the overall structural behaviour. The results of SLaMA are compared to refined numerical pushover analyses carried out with the FEM software Ruaumoko (Carr, 2016). The adopted modelling strategy, previously validated on experimental results (Magenes and Pampanin, 2004), is based on a lumped plasticity approach and is resumed in Table 7 and Figure 10. The characterisation of the structural members

is consistent to the approach adopted for SLaMA. It is worth mentioning that P-Delta effects are not modelled, given that the building is just two-storey high and made by RC. Fully fixed boundary conditions are considered at the base and floor diaphragms are modelled as rigid in plane. A uniform force profile is adopted.

**Table 7 Numerical modelling strategy.**

|  | Modelling approach | Mechanical characterisation model | Notes |
|---|---|---|---|
| Beams | Mono-dimensional Giberson elements (Sharpe, 1976) with Moment-Curvature characterisation of end sections | Moment-Curvature analysis and 50% increase in negative moment capacity due to flange effect (NZSEE 2017) | Software CUMBIA (Montejo and Priestley, 2007, Gentile, 2017) for Moment-Curvature |
| Columns | Mono-dimensional Giberson elements (Sharpe, 1976) with Moment-Axial load characterisation of end sections | Moment-Axial load interaction diagram analysis | Software CUMBIA for Moment-Axial load |
| Joints | Rigid ends in the beam-column intersections which are connected with non-linear moment-rotation springs | Behaviour of the springs follows the Equivalent Column Moment vs Drift curve (NZSEE 2017) | Drift limits for joint panels based on exp. Tests (NZSEE 2017) |

**Figure 10 Numerical modelling strategy (from Gentile et al., 2018a).**

Figure 11 shows the results of the non-linear static analyses. Firstly, the fundamental period of such frames in the transverse and longitudinal directions are equal to 0.55s and 0.47s for the Pre-2012 archetype, and 0.5s and 0.44s for the Post-2012 one. The plastic mechanism (numerically-based) of the Pre-2012 archetype frames is very similar to the one related to the Post-2012 ones, in both the transverse and longitudinal directions. In the transverse direction, the plastic mechanism, calculated at the onset of the Extensive Damage State (DS3) is characterised by the development of plastic hinges for the roof beams and base columns, and joint shear hinges at the first storey (Mixed-Sway mechanism). No brittle shear failure is registered for beams and columns (both considering SLaMA and the pushover analysis). The beam causing DS2 (yielding) is highlighted in a blue circle in Figure 11a, while the joint panel causing DS3 is highlighted with a red circle. In the longitudinal direction, the DS3 plastic mechanism is characterised by a soft-storey at the first storey, with one internal column causing DS2, but the DS3 displacement of the structure is limited by the ultimate drift in the beam-column joint highlighted with a red circle in Figure

11b,d. According to the similarities in the plastic mechanism, the capacity curves for the Pre-2012 and Post-2012 archetype frames are particularly consistent. It is worth mentioning that, the DS3 base shear for the Post-2012 archetype building is 16% and 19% greater than the Pre-2012 archetype, respectively in the transverse and longitudinal directions.

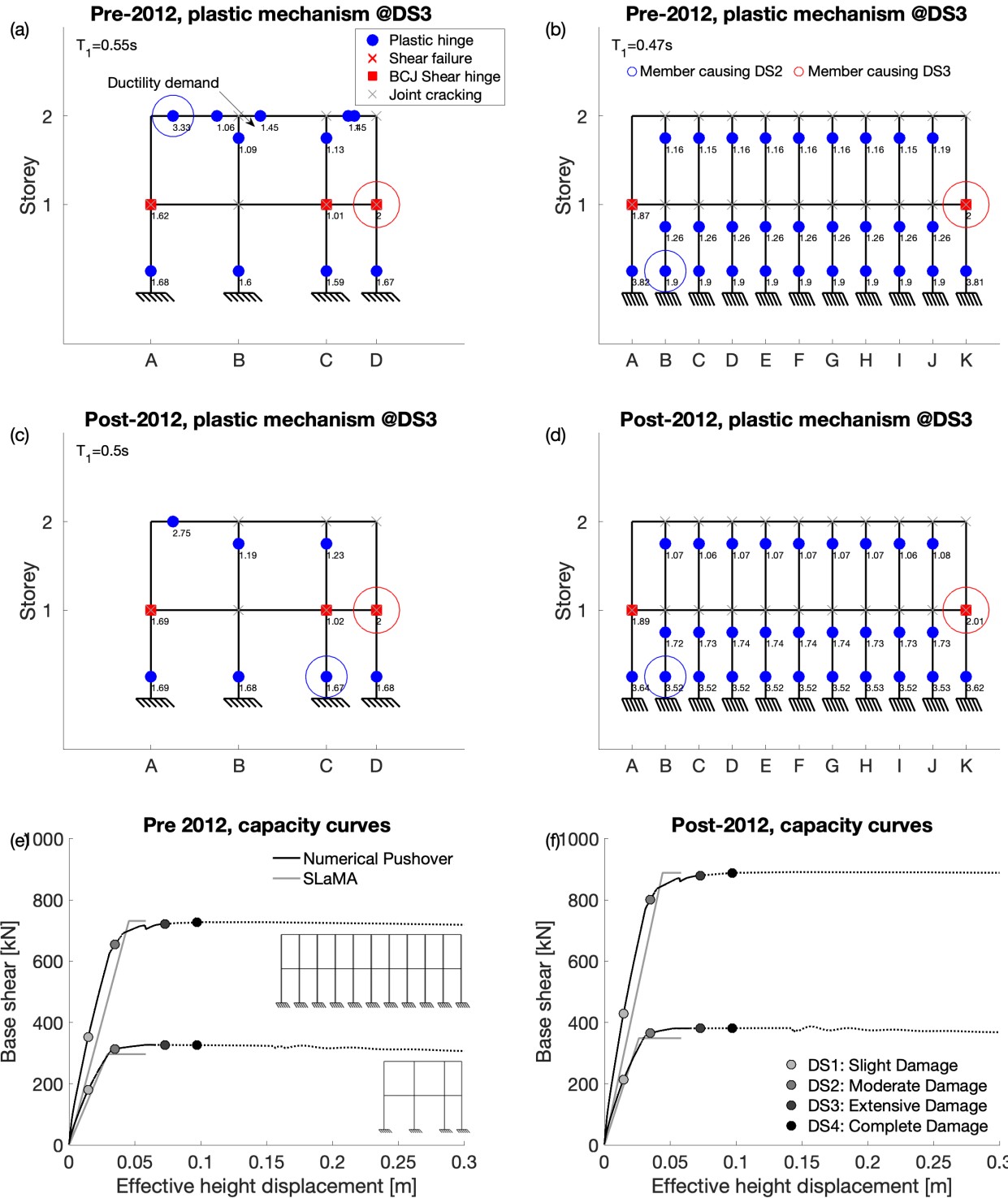

**Figure 11 Non-linear static analyses results for the archetype building.**

On the other hand, negligible differences are recorded for the displacement at DS3, for both the Pre- and Post-2012 archetypes and both transverse and longitudinal directions. This is because, in all four cases, the failure of one joint panel causes the attainment of such damage state. The displacement in the force-displacement curves is calculated at the effective height (Priestley et al., 2007), which is approximately equal to 5.50m. It is evident that the structure has a particularly-low displacement capacity, since this is limited by the low drift capacity of the joint panels. Finally, there is a satisfactory agreement between SLaMA and the numerically-based pushover, especially considering that the ultimate displacement in the SLaMA curves represents DS3.

The same two-dimensional models are analysed by means of a Cloud Analysis (Jalayer and Cornell, 2009), which consists in a series of non-linear time history analyses considering a large database of unscaled ground-motion records. For this application, the SIMBAD database (Selected Input Motions for displacement-Based Assessment and Design, Smerzini and Paolucci, 2013, Smerzini et al., 2014) is adopted. SIMBAD includes a 467 tri-axial accelerograms, generated by 130 worldwide seismic events (shallow crustal earthquakes with moment magnitudes ranging from 5 to 7.3 and epicentral distances ranging from 6 35 km). A subset of 150 records is considered here to provide a statistically significant number of strong-motion records of engineering relevance for the applications presented in this paper. As in Rossetto et al., 2016, these records are selected by first ranking the 467 records in terms of their PGA values (by using the geometric mean of the two horizontal components) and then (arbitrarily) keeping the component with the largest PGA value (for the 150 stations with highest mean PGA). This record selection strategy is compatible with the adopted cloud analysis procedure and with the lack of specific, freely-available, ground-motion databases for the considered case-study region. A rigorous site-specific, hazard-consistent record selection could be used for alternative non-linear demand estimation methods for probability-based seismic risk assessment (Jalayer and Cornell, 2009), such as multiple-stripe analysis. The models used for the numerical pushover analysis are also adopted for the cloud analysis. The hysteretic behaviour of beams and columns is characterised be the revised Takeda model (Saiidi and Sozen, 1979), with the columns having a thinner loop. On the other hand, the hysteretic behaviour of the beam-column joints is modelled using the Modified Sina model (Saiidi and Sozen, 1979), which is able to capture their pinching behaviour.

The results of the dynamic analyses (150 non-linear time history simulation for each of the four models), are used to plot a cloud of points in the plane inter-storey drift (chosen as Engineering Demand Parameter, EDP) vs pseudo-spectral acceleration at the first fundamental period ($T_1$) of each frame and for a 5% damping, i.e., $S_a(T_1)$, chosen as Intensity Measure, IM. The linear least square method is applied on those pairs in order to estimate the conditional mean and standard deviation of EDP given IM and derive the commonly-used power-law model $EDP = aIM^b$, where $a$ and $b$ are the parameters of the regression. The derived probabilistic seismic demand model is used to define a set of four fragility curves, one for each DS. Such curves are represented in the form of Eq. 2, but using $S_a(T_1)$ as intensity measure of the earthquake intensity. An average fundamental period (given the actual periods in Figure 11), equal to 0.5s, is chosen as a representative period of the considering building class. Although this might cause a small decrease in the efficiency of the IM (e.g., Minas and Galasso, 2018), such choice allows for a comparison between the different fragility curves for different buildings. This curve represents the probability of exceeding a given threshold of inter-storey drift threshold (corresponding to a given DS), conditioned on a value of the

earthquake IM. Figure 12 represents the fragility curves for the Pre-2012 and Post-2012 archetype structures, both for the longitudinal and transverse directions. The adopted inter-storey drift thresholds are defined according to the definitions in Kircher et al., 2006 by post processing the results of the pushover analyses. Those are equal to 0.25%, 0.6%, 1.25%, and 1.67% respectively for DS1, DS2, DS3, and DS4. Such values are consistent with the highlighted displacements in Figure 11, and respectively correspond to the cracking in the first member in the system, the full onset of the plastic mechanism, the attainment of 75% and 100% of the ultimate drift in the first member.

The results show that the seismic fragility of the Post-2012 structures is reduced with respect to the Pre-2012 ones (Table 8). On average, the fragility median of the Post-2012 frames are 5.6% and 4.3% higher than the Pre-2012 ones, respectively in the transverse and longitudinal directions. The related dispersion is reduced, on average, by 16.1% and 14.3% for transverse and longitudinal directions.

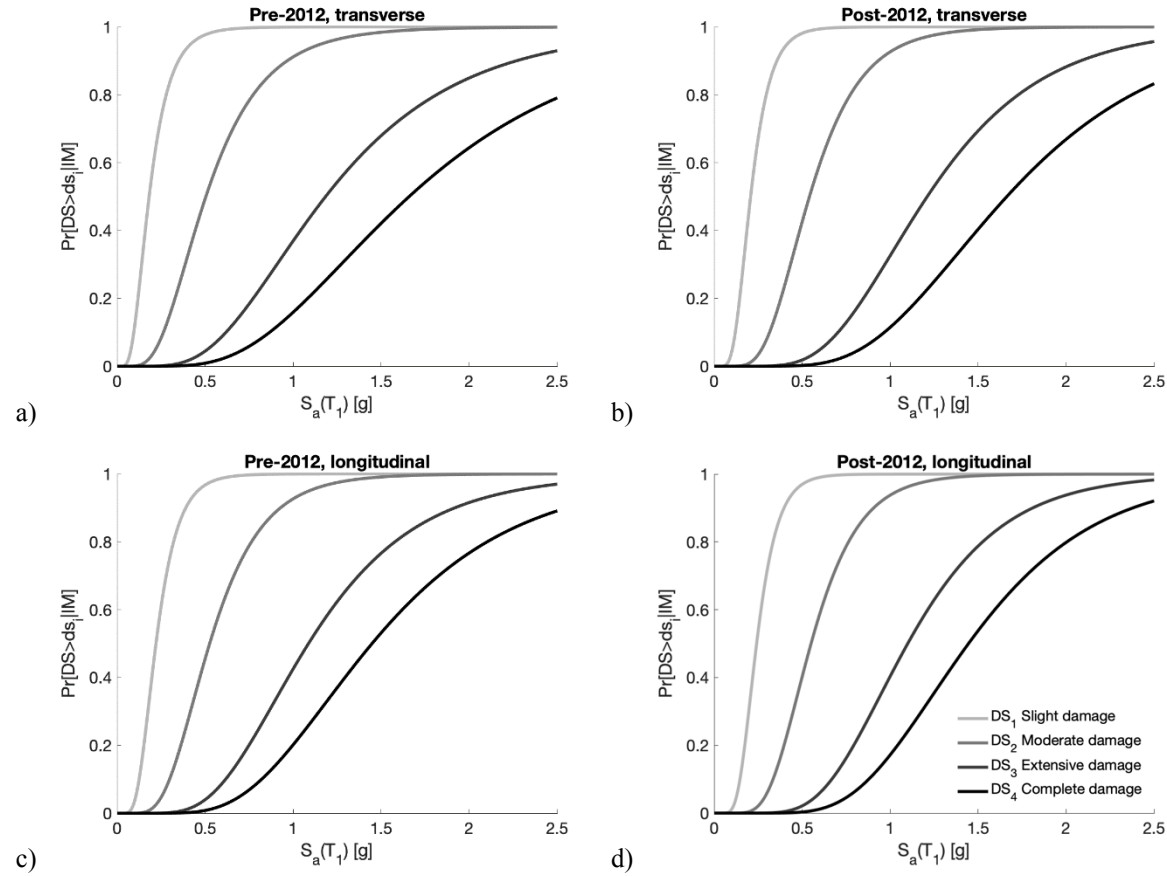

**Figure 12 Archetype building: fragility curves based on cloud analysis, calculated for Sa(0.5s).**

**Table 8 Fragility curves of the archetype frames based on cloud analysis (T₁=0.5s).**

| | | Pre-2012 | | Post-2012 | |
|---|---|---|---|---|---|
| | | $\mu$: Median $S_a(T_1)$ [g] | $\beta$: Dispersion | $\mu$: Median $S_a(T_1)$ [g] | $\beta$: Dispersion |
| Transverse | DS1 | 0.1823 | 0.5066 | 0.2078 | 0.4226 |

| | | | | | |
|---|---|---|---|---|---|
| | DS2 | 0.5045 | 0.5066 | 0.5420 | 0.4226 |
| | DS3 | 1.1855 | 0.5066 | 1.2110 | 0.4226 |
| | DS4 | 1.6588 | 0.5066 | 1.6628 | 0.4226 |
| Longitudinal | DS1 | 0.2213 | 0.4429 | 0.2412 | 0.3885 |
| | DS2 | 0.5260 | 0.4429 | 0.5506 | 0.3885 |
| | DS3 | 1.0873 | 0.4429 | 1.1000 | 0.3885 |
| | DS4 | 1.4428 | 0.4429 | 1.4552 | 0.3885 |

It is worth noting that the structure-specific fragilities obtained with the analyses presented herein could be used to, in principle, to redefine the INSPIRE index. However, this would require the derivation of numerical fragility curves for a much larger set of archetype buildings, consistently with the building typologies identified in Table 1. However, adopting the refined fragilities

as an input for the INSPIRE index definition would be inappropriate. Indeed, the proposed index is defined as a quick and practical tool for seismic vulnerability prioritisation of large building portfolios (level 1 analysis). A more time-consuming fragility analysis (level 2) should instead be used to derive quantitative seismic risk estimates for one or more selected building in the database and to design structure-specific risk mitigation strategies (e.g., structural retrofitting).

## 5. Concluding remarks

This paper introduces the INSPIRE index, which is an empirical proxy for the relative seismic risk of buildings within a given building portfolio and allows to define prioritisation schemes for risk mitigation measures. The definition of such index represents the first step of the wider framework of the *INdonesia School Programme to Increase REsilience* (INSPIRE), which aims to develop an advanced, harmonised and science-based risk assessment framework for school infrastructure in Indonesia subjected to cascading earthquake-tsunami hazards.

To this aim, a data collection form used for the rapid visual inspection of RC buildings is first developed and presented. Such form allows to calculate the INSPIRE seismic risk prioritisation index, the Tsunami PTVA index (in any of its variations), to obtain a level of geometrical/mechanical information sufficient to define one or more archetype buildings (representative of the portfolio) and/or to build refined numerical models, provided that simulated design is adopted to cross check the available information.

The INSPIRE index is specifically calibrated for RC buildings, and consists of two parts: a baseline score and a performance modifier. The baseline score is based on the HAZUS MH4 fragility curves, while the performance modifier is based on the score of the building with regard to eight "secondary" parameters, which, if present, are deemed to jeopardise the building performance. To minimise subjectivity, the relative weight of the secondary parameters is defined according to the Analytic Hierarchy Process. This allows to have a rational and mathematically-consistent assignment of the weights which is based on

pairwise comparisons between the secondary parameters and eigenvalues theory.

The INSPIRE form and seismic risk prioritisation index are adopted for the analysis of 85 RC school buildings in the city of Banda Aceh, Indonesia, which is located in the Sumatra Island, the area mostly affected by the 2004 Indian Ocean earthquake-

tsunami sequence. The joint application of the INSPIRE seismic risk prioritisation index and the PTVA tsunami index allow to define a clear and transparent rationale behind any prioritisation schemes for school buildings in Banda Aceh. In fact, the relative seismic risk of the considered buildings is particularly similar, while the relative tsunami risk shows a strong dependence with the distance from the coast. Indeed, the results show that a multi-hazard-based priority list is mostly governed

by the tsunami risk for the case-study portfolio.

The advantages of using the INSPIRE form are further demonstrated by defining two archetype buildings, representative of the portfolio, based on the RVS results. The seismic performance of the archetype buildings are firstly analysed by means of non-linear static analyses, both analytically using the Simple Lateral Mechanism Analysis (SLaMA), and numerically using refined finite-element models. Finally, the archetypes are analysed by means of cloud analysis, performing non-linear dynamic

analyses using 150 unscaled natural ground motions and deriving fragility curves.

The results in this paper demonstrate the effectiveness of both the INSPIRE RVS form and INSPIRE seismic risk prioritisation index in providing a rational method to derive a prioritisation scheme, which can be extended including multi-hazard considerations, and in allowing the definition of an archetype building for more detailed evaluations/analyses.

This study represents a first step toward a comprehensive framework for earthquake and tsunami vulnerability and risk

assessment and the selection of optimal retrofitting strategies for school facilities in Indonesia, through a to a multi-criteria decision-making analysis. Future research will investigate the numerically-based tsunami fragility of the archetype buildings adopting different approaches (e.g., Petrone et al., 2017) and a full seismic loss analysis considering non-structural components, which often represent the highest share of the seismic losses.

**Acknowledgements**

This study was performed in the framework of the "*INSPIRE: INdonesia School Programme to Increase REsilience*" and "*i-RESIST: Increasing REsilience of Schools in Indonesia to earthquake Shaking and Tsunami*" projects, funded by the British Council through the Newton Institutional Links scheme and Research England through the University College London (UCL) Global Challenges Research Fund (GCRF) Small Research Grants scheme.

The Tsunami and Disaster Mitigation Research Centre (TDMRC) at Syiah Kuala University is acknowledged for the technical

support during the field work. Prof Peter Sammonds, Mr Karim Aljawhari, Dr Nurmalahayati Nurdin from UCL are gratefully acknowledged for helping planning and executing the data collection. Mr Rifqi Irvansyah, Ms Putri Istiqamah, Ms Sausan Zahrah, Ms Natasya Amalia, Mr Fajarul Aulia, Mr Ilham M Siddiq are gratefully acknowledged for their help in the data collection.

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
