# Peer review of "From rapid visual survey to multi-hazard risk prioritisation and numerical fragility of school buildings"

_Natural Hazards and Earth System Sciences, 2018_

## Referee Comment (RC1) · Anonymous Referee #1 · 23 Jan 2019

I feel that this manuscript needs to improve their structure, connection between sections (especially the last section on fragility functions), add some explanations and proper references. General comments 1. I think the authors should better show uniqueness of their rapid survey form, i.e. how their new rapid survey form differs to other rapid survey forms, easier/faster to fill?, can be used for various purposes, etc. 2. Is INSPIRE developed mainly for earthquake and tsunami or applicable to other hazards? If the later, more explanations are needed as only examples on earthquake and tsunami were demonstrated. 3. The newest PTVA is PTVA4 that calibrated their

vulnerability based on comments and questionnaire results from experts in this field. Why don't you use the newest one? Reference: Dall'Osso, F., Dominey-Howes, D., Tarbotton, C., Summerhayes, S., and Withycombe, G.: Revision and improvement of the PTVA-3 model for assessing tsunami building vulnerability using "international expert judgment": introducing the PTVA-4 model, Nat. Hazards, 83, 1229–1256,2016. Izquierdo, T., Fritis, E., and Abad, M.: Analysis and validation of the PTVA tsunami building vulnerability model using the 2015 Chile post-tsunami damage data in Coquimbo and La Serena cities, Nat. Hazards Earth Syst. Sci., 18, 1703-1716, 2018. Alternatively, you can also use or compare with previously developed tsunami fragility functions of RC buildings. Reference: Suppasri, A., Charvet, I., Imai, K. and Imamura, F. (2015) Fragility curves based on data from the 2011 Great East Japan tsunami in Ishinomaki city with discussion of parameters influencing building damage, Earthquake Spectra, 31 (2), 841-868.

Specific comments 1. P2 L3: Needs reference 2. Introduction section shall be rearranged for better readability. For example, grouping the literature reviews to RC building, school building and structure of INSPIRE. At present, explanations of methods and objectives are mixed up, please rearrange and make it clear (page 3). 3. P9 L1-2: Calibrate the baseline score to what? Why DS3 is used? 4. P10 Table 3: How these weight factors obtained? If from HAZUS, how certain these values can be applied globally? 5. P12 Section 3.3: If there are three hazards or more, how equation 7 and Fig. 3 will be? And how to avoid such double count of subsequent damage? 6. P14 L31: Needs reference 7. P15 and/or P19: I think you should give more explanations about tsunami hazard in your study area in the past/future. How the flow depth of the 2004 Indian Ocean tsunami used in INSPIRE. I am not sure if they have measured flow depth in all buildings in your study if so, the flow depths are from model simulation? 8. P20 Fig. 8 There should be some discussions that point out importance of considering multi-hazard scenarios. For example, buildings that became higher risk when tsunami is considered and comments on how the developed map can be used for disaster planning. 8. P16 Fig. 5: Add photo taken dates 9. P21 Section 4.3: I feel

that this section is not related to others otherwise, it should be used to compare with analysis results of other previous sections. What was the purpose of this section? Why didnt the authors use their own developed fragility functions instead of HAZUS?
* * *

---

## Referee Comment (RC2) · Anonymous Referee #2 · 23 Feb 2019

General Comments: This manuscript provides a timely discussion on how to accomplish strategic prioritisation of intervention on school buildings in a transparent way using the Analytic Hierarchy Process in a multi-hazard context (earthquakes and tsunami). The approach is validated though a detailed analysis and a simplified mechanical methodology (i.e., SLAMA).

I report in the following what I consider minor comments that could improve the overall quality of the manuscript.

1) The explicit reference to Banda Aceh in the title could be removed as the methodology and approach is rather general and the case of Indonesia is a case-study 2) More discussion on the problem of "code enforcement" should be provided. The approach of classifying building according to the release of building codes is rationale, it makes sense, it refers to a widespread practice in regional analyses but a comparison with the real construction practice should be provided. In the specific case this is possible (e.g., comparison of reinforcement in figure 5e with code provisions. 3) I personally do not agree with the low weight given to soil conditions in the matrix A. Is the case study area located in a relatively firm soil area? A comparison with the Vs30 model based on slope from USGS should be provided and discussed (Allen and Wald 2009) 4) It is not clear how and if brittle failures are accounted for in the detailed procedure presented in the second part of the work. If not, a simple approach for element classification as ductile or brittle could be attempted comparing the amount of longitudinal and transversal reinforcement ration in typical elements as done in some previous work for the L'Aquila case in Italy (De Luca and Verderame 2013).

Specific Comments:

Page 1 Line 10 – change Resilience with REsilience to be consistent with the acronym Line 17 – change demonstrated with implemented on Line 30 – add a comma after the closed bracket Page 2 Line 10 – Some of the references to prioritisation programme of schools in other countries should be already cited here Line 33 – the importance of schools should be discussed including a reference form UNHDR or UN. Page 3 Line 1 – the use of schools as shelters in crisis is highly debated can you add a reference on this topic to acknowledge this aspect? Line 14 – change representative with consistent with building codes and practice of the country. Page 5 Line 33 – it should be mechanism-based and not mechanics-based Page 6 Line 21 – on what basis DS3 is considered equivalent to life-safety, are you basing this on Hazus, EMS98 etc. Further specification on this is necessary Page 8 Table 2 – This table is too dense, try to reduce/condense the text in this table Page 9 Line 13 – Pmax and Pmin in the equation

are those indicated in Fig 2a or in Fig 2b? I assumed it is Fig 2a, if this is the case, I would remove the grey dots in Figure 2b. Page 10 Line 2 – On what basis you assumed 25%? Do you have a reference or any evidence for this assumption? Table 3 – What is the rationale for the weights? Why unfavourable soil is so low? See general comment 3) Page 11 Table 4 – is a scoring system from 1 to 9 too granular as it is based on expert judgement? Table 5 – again why unfavourable soil is so low Page 12 Line 24-27 – Asprone et al. used a similar multi-hazard index in 2013, compare differences with this approach. Page 13 Line 12-16 – The 50-50 split should be assumed and changes on the basis of how suitable are Hazus typologies with respect to the building stock to the country considered. In a more general context this could be 70-30 or 30-70 if the typologies are more or less representative of the building stock. I understand this is arbitrary, but more discussion should be provided on this. Page 16 Line 10 – was there any double-check of code-enforcement? Situations like Figure 5e allow this sort of discussion and this should be provided. See general comment 2). Page 17 Line 6 – why you assumed modal values (and not median for example?) Page 18 Line 9 – Are you referred to length of the elements or section dimensions? If this is the overall length of the elements why they increased with time? Page 19 Line 1 – again, did they correspond to what was prescribed by code? Table 6 – I am surprised that 2012 code was not prescribing stirrups in joints, is there again a difference between practice and code? Page 23 Line 14 – can SLAMA account for brittle failures? If not a preliminary classification of the elements as ductile or brittle could be useful, see general comment 4). Page 25 Line 26 – can you provide a reference for the drift thresholds? Page 27 Line 26-28 – How did you compared the Inspire index results with results of the fragilities? A more detailed discussion should be provided right after Table 8. At the moment the comparison/validation is not very clear.

References:

Allen, T. I., & Wald, D. J. (2009). On the use of high-resolution topographic data as a proxy for seismic site conditions (VS 30). Bulletin of the Seismological Society of

America, 99(2A), 935-943.

Asprone, D., De Risi, R., & Manfredi, G. (2016). Defining structural robustness under seismic and simultaneous actions: an application to precast RC buildings. Bulletin of Earthquake Engineering, 14(2), 485-499.

De Luca, F., & Verderame, G. M. (2013). A practice-oriented approach for the assessment of brittle failures in existing reinforced concrete elements. Engineering Structures, 48, 373-388.

---

## Author Comment (AC1) · 4 Apr 2019

**Roberto Gentile and Carmine Galasso**

r.gentile@ucl.ac.uk

Received and published: 4 April 2019

Legend: blue - reviewer's comment. Black - answers from the authors.

I feel that this manuscript needs to improve their structure, connection between sections (especially the last section on fragility functions), add some explanations and proper references.

We would like to thank this reviewer for the valuable comments on our manuscript. We will improve the manuscript structure and connection between sections accordingly.

General Comments I think the authors should better show uniqueness of their rapid survey form, i.e. how their new rapid survey form differs to other rapid survey forms, easier/faster to fill?, can be used for various purposes, etc.

This reviewer is acknowledged for this comment. In fact, although the main features of the INSPIRE Rapid Visual Survey (RVS) are discussed in the manuscript, the comparison with other approaches and its uniqueness will be better stressed-out in the revised version of the paper. The unique features of the INSPIRE form are related to: 1) the possibility to calculate both a seismic and a tsunami index while requiring a reasonable amount of time to be filled; 2) the consideration of qualitative confidence levels for each parameter, which is particularly useful in deriving statistics, defining archetype buildings and/or numerical models (as shown in the paper); 3) with simple customisations, it can be used for other purposes (e.g., considering other types of hazards).

Is INSPIRE developed mainly for earthquake and tsunami or applicable to other hazards? If the later, more explanations are needed as only examples on earthquake and tsunami were demonstrated. Currently, the INSPIRE RVS form is optimised for earthquake and tsunami. This is why the specific case study in the illustrative application (i.e., Banda Aceh) is mainly affected by such hazards. However, simple modifications of the form could be easily implemented to include other hazards (e.g., more information related to roof type/quality/connections is required when dealing with wind vulnerability). This concept will be emphasised in the revised manuscript. It is worth mentioning, however, that the multi-hazard considerations presented in Section 3.3 are still valid, and are not limited to the earthquake and tsunami hazards.

The newest PTVA is PTVA4 that calibrated their vulnerability based on comments and questionnaire results from experts in this field. Why don't you use the newest one? Reference: Dall'Osso, F., Dominey-Howes, D., Tarbotton, C., Summerhayes, S., and Withycombe, G.: Revision and improvement of the PTVA-3 model for assessing tsunami building vulnerability using "international expert judgment": introducing the PTVA-4 model, Nat. Hazards, 83, 1229–1256,2016. Izquierdo, T., Fritis, E., and Abad,

**NHESSD**
*M.:* Analysis and validation of the PTVA tsunami building vulnerability model using the 2015 Chile post-tsunami damage data in Coquimbo and La Serena cities, Nat. Hazards Earth Syst. Sci., 18, 1703-1716, 2018. Alternatively, you can also use or compare with previously developed tsunami fragility functions of RC buildings. Reference: Suppasri, A., Charvet, I., Imai, K. and Imamura, F. (2015) Fragility curves based on data from the 2011 Great East Japan tsunami in Ishinomaki city with discussion of parameters influencing building damage, Earthquake Spectra, 31 (2), 841-868. The reviewer is particularly acknowledged for this comment. In the revised manuscript, the newly-calibrated PTVA4 index will be used. In the figures below, the results of the PTVA4 index (right column) for the analysed building portfolio are compared to the previously-adopted PTVA3 index (left column). The resulting vulnerability index for some of the buildings has indeed slightly changed when the new methodology is applied. However, it is worth highlighting that using the new calibration (PTVA4) has limited-to-negligible effects on the overall multi-hazard prioritisation for the considered portfolio.

*Specific comments P2 L3: Needs reference* The reference (last Italian census of 2011) will be added in the revised manuscript:

Istituto nazionale di STATistica, ISTAT (2011). 15th general census of population and housing (in Italian). http://dati-censimentopopolazione.istat.it.

Introduction section shall be rearranged for better readability. For example, grouping the literature reviews to RC building, school building and structure of INSPIRE. At present, explanations of methods and objectives are mixed up, please rearrange and make it clear (page 3). To improve the readability of the paper, the literature review in the introduction will be rearranged as suggested from the reviewer.

*P9 L1-2: Calibrate the baseline score to what? Why DS3 is used?* The reviewer is acknowledged for this comment. Firstly, the correct wording in this case should be "to define" rather than "to calibrate" the baseline score, and this change will be implemented in the revised manuscript. Moreover, as discussed in Section 3.1, the definition of the
baseline score of the index is based on the DS3 damage state, as defined in HAZUS. This is because DS3 is deemed to be related to the Life-Safety performance objective in modern seismic codes. To further expand on this, when considering an RC member within a frame (e.g., beam or column), DS3 corresponds to the member ultimate capacity, which can be related to flexural failure (ultimate strain in concrete or steel, buckling of the reinforcement), shear or lap-splice failure. According to modern seismic codes (e.g. NZSEE 2017, ASCE 41-13, EuroCode 8), such a damage condition (for one or a few members) would define the ultimate limit state of the frame, which is "conventionally" related to the safety of people occupying the structure (i.e., Life-Safety). Such more-detailed comments will be implemented in the revised manuscript.

NZSEE: New Zealand Society for Earthquake Engineering, The seismic assessment of existing buildings - technical guidelines for engineering assessments. Wellington, New Zealand, 2017.

ASCE 41-13 (2014), Seismic Evaluation and Retrofit of Existing Buildings, American Society of Civil Engineer and Structural Engineering Institute, Reston, Virginia, USA.

EC8 (2005), 'European Comittee for Standardisation. Eurocode 8: Design of structures for earth- quake resistance. Part 3: Strengthening and repair of buildings'.

*P10 Table 3: How these weight factors obtained? If from HAZUS, how certain these values can be applied globally?* The secondary parameters are selected to account for aspects of the analysed building which are not-explicitly considered in the HAZUS framework (i.e., in the baseline score). Such parameters, if present in the building, are deemed to negatively-affect its seismic performance. As mentioned in Section 3.2, the weights for such parameters (representing their relative importance) are defined according to the analytic hierarchy process, in turn based on all the possible pairwise comparisons between such parameters. At the present stage, the expert judgement that defines the pairwise comparisons is provided by the authors. However, in the future such coefficients will be updated considering the opinion of a group of experts in the

NHESSD
field of structural and earthquake engineering. Finally, the weights of the secondary parameters are portfolio-specific, and therefore they should be calibrated differently for each analysis situation. Such concepts will be further discussed in the revised manuscript.

*P12 Section 3.3: If there are three hazards or more, how equation 7 and Fig. 3 will be? And how to avoid such double count of subsequent damage?* This comment is particularly acknowledged. Equation 7 (Section 3.3) is defined for an arbitrary number of dimensions k. A high-dimensional Euclidean space is considered in which each dimension represents the vulnerability/risk index for one hazard. A building subjected to k different hazards will be defined as a point in this space. It is proposed to define a multi-hazard index as the distance of this point from the origin. The second part of the comment relates to subsequent/cumulative damage. It worth mentioning that considering cumulative damage related to the different hazards is outside the scope of this prioritisation scheme. Such important concept will be discussed in the revised manuscript.

P14 L31: Needs reference The reference will be added in the revised manuscript.

Seta, W.J.,: Atlas Lengkap Indonesia dan Dunia (untuk SD, SMP, SMU, dan Umum). Pustaka Widyatama. p.7, 2000.

P15 and/or P19: I think you should give more explanations about tsunami hazard in your study area in the past/future. How the flow depth of the 2004 Indian Ocean tsunami used in INSPIRE. I am not sure if they have measured flow depth in all buildings in your study if so, the flow depths are from model simulation? This comment is particularly appreciated. The tsunami height (relative to the ground) related to the 2004 Indian Ocean event is based on the field-measures by lemura et al., 2012. In this work, a correlation model between the distance from the coast and the tsunami height was developed. Such a model is adopted here to calculate, for each building in the portfolio, the expected tsunami height as a function of the distance from the coast.
Such considerations will be added in Section 4.2 of the revised paper.

Iemura, H., Pradono, M.H., Sugimoto, M., Takahashi, Y. and Husen, A.: Tsunami height memorial poles in banda aceh and recommendations for disaster prevention. Proceedings of the International Symposium on Engineering Lessons Learned from the 2011 Great East Japan Earthquake, March 1-4, 2012, Tokyo, Japan, 2012.

P20 Fig. 8 There should be some discussions that point out importance of considering multi-hazard scenarios. For example, buildings that became higher risk when tsunami is considered and comments on how the developed map can be used for disaster planning. We thank the reviewer for this comment and agree on the importance of considering multi-hazard scenarios for risk prioritization. Indeed, a specific comment on this aspect will be added after Figure 8. Furthermore, a comment on how these maps can be used for disaster planning will be added. For instance, the developed maps could be used to identify "safer areas" where strategic buildings (e.g., schools or hospitals) should be located. While specific recommendations on disaster planning are outside the scope of this study, a few references to this aspect will be included in the revised paper:

Alexander D.: Disaster and Emergency Planning for Preparedness, Response, and Recovery, Oxford Research Encyclopedia of Natural Hazard Science, doi: 10.1093/acrefore/9780199389407.013.12, 2019.

*P16 Fig. 5: Add photo taken dates* These pictures have been taken during the field-work carried out between 16 and 19 October 2018. This information will be added in the revised manuscript.

P21 Section 4.3: I feel that this section is not related to others otherwise, it should be used to compare with analysis results of other previous sections. What was the purpose of this section? Why didnt the authors use their own developed fragility functions instead of HAZUS? As mentioned in the introduction of the paper, the ISPIRE form has multiple purposes, allowing to carry out analyses with different levels of reInteractive comment
finement. Clearly, a prioritisation scheme can be defined according to the relative risk index. Moreover, the form provides enough data to build refined numerical models for one or more selected buildings (e.g., an "average" archetype building or the buildings with the highest relative risk index). The purpose of Sections 4.2 and 4.3 is respectively to illustrate the above-mentioned purposes of the form. Accordingly, a more clear aim for Section 4.3 and a better link to the previous sections will be added in the revised manuscript.

In this context, the HAZUS-based fragilities are not used to derive quantitative estimates of the seismic risk of one or more specific building in the database, but only to define relative estimates (i.e., the prioritisation scheme). Conversely, the structurespecific fragilities obtained with the analyses in Section 4.3 can be used to provide such quantitative estimates of the seismic risk. Therefore, comparing the HAZUSbased fragility curves with the refined ones derived in Section 4.3 is deemed to be inappropriate.

Finally, adopting the refined fragilities as an input for the INSPIRE index would have the opposite of the desired effect. The index is defined as a quick and practical estimation tool for large portfolios (level 1). A time-consuming fragility estimation (level 2) should not be used as an input to a quick relative-risk method.

Please also note the supplement to this comment:

https://www.nat-hazards-earth-syst-sci-discuss.net/nhess-2018-397/nhess-2018-397-AC1-supplement.pdf

**NHESSD**
**NHESSD**

Fig. 1: comparison of the prioritisation based on the PTVA3 and PTVA4 indices.

---

## Author Comment (AC2) · 4 Apr 2019

Legend: blue – reviewer's comment. Black – answers from the authors.

General Comments

This manuscript provides a timely discussion on how to accomplish strategic prioritisation of intervention on school buildings in a transparent way using the Analytic Hierarchy Process in a multi-hazard context (earthquakes and tsunami). The approach is validated though a detailed analysis and a simplified mechanical methodology (i.e.,

SLAMA). I report in the following what I consider minor comments that could improve the overall quality of the manuscript.

1. The explicit reference to Banda Aceh in the title could be removed as the methodology and approach is rather general and the case of Indonesia is a case-study

The reference to Banda Aceh in the title will be removed in the revised manuscript.

2. More discussion on the problem of "code enforcement" should be provided. The approach of classifying building according to the release of building codes is rationale, it makes sense, it refers to a widespread practice in regional analyses but a comparison with the real construction practice should be provided. In the specific case this is possible (e.g., comparison of reinforcement in figure 5e with code provisions.

This comment is particularly appreciated. The structural details of the archetype buildings for this study are defined based on Figure 5e (and other similar photos). Indeed, the simulated design approach according to the two considered seismic codes is adopted to confirm such visually-based assumptions. As it will be stressed out in the revised manuscript, the amount of longitudinal reinforcement observed in the field was greater than the minimum by code. On the other hand, based on the limited visual information available for the transverse reinforcement, no joint stirrups were conservatively considered for both the Pre-2012 and Post-2012 vulnerability classes, regardless of the requirement in both codes.

3. I personally do not agree with the low weight given to soil conditions in the matrix A. Is the case study area located in a relatively firm soil area? A comparison with the Vs30 model based on slope from USGS should be provided and discussed (Allen and Wald 2009)

The authors acknowledge the comment of this reviewer. However, two considerations should be given herein. Firstly, the prioritisation will be affected by the building-to-building variability in one criterion (in this case soil type), rather than the absolute

values. For this particular building portfolio, the soil type is particularly uniform (shear wave velocity in the first 30 meters of soil in the range 150-250m/s). Such comment will be added in the revised manuscript. Secondly, as it will be emphasised in the revised paper, the weights of the secondary parameters should in theory be portfolio-specific, and therefore calibrated differently for each analysis situation.

4. It is not clear how and if brittle failures are accounted for in the detailed procedure presented in the second part of the work. If not, a simple approach for element classification as ductile or brittle could be attempted comparing the amount of longitudinal and transversal reinforcement ration in typical elements as done in some previous work for the L'Aquila case in Italy (De Luca and Verderame 2013).

This comment is particularly acknowledged. Both in SLaMA and the refined numerical analyses, each beam and column in the system has been characterised considering many possible failure mechanisms (i.e., flexure, bar buckling, lap-splice failure, shear), considering that the weakest will govern its behaviour. For this particular case study, the plastic mechanisms of the archetype buildings are characterised by first shear cracking and/or shear failure in the joints and flexural plastic hinges in beams and columns (Figure 11a,b,c,d). No brittle failure is registered for beams and columns. This point will be stressed out in the revised manuscript.

Specific Comments

5. Page 1 Line 10 – change Resilience with REsilience to be consistent with the acronym; Line 17 – change demonstrated with implemented on Line 30 – add a comma after the closed bracket

These editorial changes will be implemented.

6. Page 2 Line 10 – Some of the references to prioritisation programme of schools in other countries should be already cited here Line 33 – the importance of schools should be discussed including a reference form UNHDR or UN.

Three of the references in Section 2 will be moved to the introduction (page 2, line 10) as suggested by the reviewer. Moreover, a specific reference to the UN campaigns for world disaster risk reduction will be provided.

United Nations Centre for Regional Development, UNCRD,: Reducing vulnerability of school children to earthquakes. UNCDR report, 2009.

7. Page 3 Line 1 – the use of schools as shelters in crisis is highly debated can you add a reference on this topic to acknowledge this aspect? Line 14 – change representative with consistent with building codes and practice of the country.

This comment is acknowledged. According to UN, educational continuity should be prioritised in disaster conditions. Therefore, the reference to schools adopted as a shelter will be removed in the revised manuscript. Regarding the second comment, we will change representative with consistent with building codes and practice of the country.

8. Page 5 Line 33 – it should be mechanism-based and not mechanics-based

This editorial change will be implemented.

9. Page 6 Line 21 – on what basis DS3 is considered equivalent to life-safety, are you basing this on Hazus, EMS98 etc. Further specification on this is necessary

This comment is particularly acknowledged. As discussed in Section 3.1, the definition of the baseline score of the index is based on the DS3 damage state, as defined in HAZUS. This is because DS3 is deemed to be connected to the Life-Safety performance objective in modern seismic codes. To further expand on this, when considering an RC member within a frame (e.g., beam or column), DS3 corresponds to the member ultimate capacity, which can be related to flexural failure (ultimate strain in concrete or steel, buckling of the reinforcement), shear or lap-splice failure. According to modern seismic codes (e.g. NZSEE 2017, ASCE 41-13, EuroCode 8), such a damage condition (for one or a few members) would define the ultimate limit state of the frame, which

is "conventionally" related to the safety of people occupying the structure (i.e., Life-Safety). Such more-detailed comments will be implemented in the revised manuscript.

NZSEE: New Zealand Society for Earthquake Engineering, The seismic assessment of existing buildings - technical guidelines for engineering assessments. Wellington, New Zealand, 2017.

ASCE 41-13 (2014), Seismic Evaluation and Retrofit of Existing Buildings, American Society of Civil Engineer and Structural Engineering Institute, Reston, Virginia, USA.

EC8 (2005), 'European Comittee for Standardisation. Eurocode 8: Design of structures for earth- quake resistance. Part 3: Strengthening and repair of buildings'.

10. Page 8 Table 2 – This table is too dense, try to reduce/condense the text in this table

An effort to reduce and condense the information in this table will be made.

11. Page 9 Line 13 – Pmax and Pmin in the equation are those indicated in Fig 2a or in Fig 2b? I assumed it is Fig 2a, if this is the case, I would remove the grey dots in Figure 2b.

$P_{HAZUS,max}$ and $P_{HAZUS,min}$ in the equation are defined based on the absolute maximum and minimum fragility values in the (selected portion of) HAZUS fragility database (Figure 2b). Therefore, the grey dots in Figure 2a will be removed in the revised manuscript.

12. Page 10 Line 2 – On what basis you assumed 25

The performance modifier is defined in the interval [0

13. Page 11 Table 4 – is a scoring system from 1 to 9 too granular as it is based on expert judgement?

Such a scoring system is the one defined in the original study/book introducing the AHP

(Saaty, 1980). It was successfully adopted in other engineering applications available in the literature (Caterino et al., 2008, Sangiorgio et al., 2019). This is the reason why it has been adopted for this study.

Saaty, T. L.: The analytical hierarchy process: Planning, priority setting, resource allocation. London: McGraw-Hill, 1980.

Caterino, N., Iervolino, I., Manfredi, G., Cosenza, E.: Multi-criteria decision making for seismic retrofitting of RC structures. Journal of Earthquake Engineering 12:555-583, 2008.

Sangiorgio, V., Pantoja, J. C., Varum, H., Uva, G., Fatiguso, F. (2019). Structural degradation assessment of RC buildings: Calibration and comparison of semeiotic-based methodology for decision support system. Journal of Performance of Constructed Facilities, 33(2)

14. Table 5 – again why unfavourable soil is so low

Please refer to the answer to comment 3.

15. Page 12 Line 24-27 – Asprone et al. used a similar multi-hazard index in 2013, compare differences with this approach.

In the work by Asprone et al., the authors propose to define a domain representing the capacity-to-demand ratio related to hazard j, as a function of the demand for a given value of the hazard i. This is defined for a quantitative analysis approach, therefore theoretically leading to a high number of analyses, especially if more than two hazards are considered. Contrarily, in the simplified approach proposed in this work it is assumed, rather than calculated, a shape to the multi-hazard domain. This comparison will be discussed in the revised manuscript.

16. Page 13 Line 12-16 – The 50-50 split should be assumed and changes on the basis of how suitable are Hazus typologies with respect to the building stock to the country considered. In a more general context this could be 70-30 or 30-70 if the typologies

are more or less representative of the building stock. I understand this is arbitrary, but more discussion should be provided on this.

The suggestion from this reviewer is rationale and valuable. However, the authors believe that such an approach would be against the definition of the INSPIRE index. Indeed, given the assumption of relying on the HAZUS database, an effort has been made to decouple, as much as possible, the aspects of the building(s) that are covered in HAZUS, represented by the baseline score, and the ones that are not covered, represented by the performance modifier. The selection of the appropriate fragility curves to apply for each building in the considered portfolio is an expert decision provided by the user. As it will be stressed out in the revised manuscript, any other type of fragility curves, if deemed appropriate, can be used to define the index.

17. Page 16 Line 10 – was there any double-check of code-enforcement? Situations like Figure 5e allow this sort of discussion and this should be provided. See general comment 2).

The visually-based information on the structural details have been compared to the results of the simulated design process. The final definition of the archetype building is based on both approaches. Please also refer to the answer to comment 2.

18. Page 17 Line 6 – why you assumed modal values (and not median for example?)

In our opinion, an archetype building, being representative of a given building class, should reflect the more frequent geometric/material characteristics observed over the entire portfolio. Such a condition is achieved when distribution modal values are adopted.

19. Page 18 Line 9 – Are you referred to length of the elements or section dimensions? If this is the overall length of the elements why they increased with time?

The authors refer to the depth of the cross-section. This editorial change will be provided in the revised manuscript.

[Figure]

20. Page 19 Line 1 – again, did they correspond to what was prescribed by code? Table 6 – I am surprised that 2012 code was not prescribing stirrups in joints, is there again a difference between practice and code?

Based on the limited visual information available for the transverse reinforcement, no joint stirrups were conservatively considered for both the Pre-2012 and Post-2012 vulnerability classes, regardless of the requirement in both codes. Please also refer to the answer to comment 2.

21. Page 23 Line 14 – can SLAMA account for brittle failures? If not a preliminary classification of the elements as ductile or brittle could be useful, see general comment 4).

For beams and columns, SLaMA is capable to consider flexural failures, as well as lap-splice failures, rebar buckling and shear failure. Therefore, brittle failures are considered in this study. Please refer to the answer to comment 4.

22. Page 25 Line 26 – can you provide a reference for the drift thresholds?

This comment is particularly appreciated. The adopted inter-storey drift thresholds are defined according to the definitions in Kircher et al., 2006 by post processing the results of the pushover analyses. Such values are consistent with the highlighted displacements in Figure 11 and are building-specific. For this reason, no further reference is needed, since such values are directly calculated in the context of this study.

Kircher C.A., Whitman R.V. and Holmes W.T.: HAZUS Earthquake Loss Estimation Methods". Natural Hazard Review, 7:45-59, 2006.

23. Page 27 Line 26-28 – How did you compared the Inspire index results with results of the fragilities? A more detailed discussion should be provided right after Table 8. At the moment the comparison/validation is not very clear.

Comparing the HAZUS-based fragility curves with the refined ones derived in section 4.3 is deemed to be inappropriate. In fact, those two types of curves have a

particularly-different purpose. The HAZUS-based fragilities, at least in this context, are used to define relative estimates of the seismic risk (i.e., the prioritisation scheme). Conversely, the structure-specific fragilities obtained with the analyses in Section 4.3 can be used to provide such quantitative estimates of the seismic risk. As it will be conveyed in the revised manuscript, the purpose of Section 4.3 is not to compare/validate the adopted HAZUS fragilities. Conversely, this section shows that using the INSPIRE form allows to define refined numerical models of some selected buildings (in this case, the archetype buildings). These concepts will be added in the revised manuscript.